# Senescent cancer-associated fibroblasts in pancreatic adenocarcinoma restrict CD8+ T cell activation and limit responsiveness to immunotherapy in mice

Benjamin Assouline[1,14], Rachel Kahn[1,14], Lutfi Hodali[1,14], Reba Condiotti[1], Yarden Engel[2], Ela Elyada[3], Tzlil Mordechai-Heyn[1,4], Jason R. Pitarresi[5,6], Dikla Atias [7], Eliana Steinberg[8], Tirza Bidany-Mizrahi[2], Esther Forkosh[9], Lior H. Katz[9], Ofra Benny [8], Talia Golan[7], Matan Hofree[2,10], Sheila A. Stewart [11], Karine A. Atlan[12], Gideon Zamir[4], Ben Z. Stanger [13], Michael Berger [2] & Ittai Ben-Porath [1] ✉

Senescent cells within tumors and their stroma exert complex pro- and anti-tumorigenic functions. However, the identities and traits of these cells, and the potential for improving cancer therapy through their targeting, remain poorly characterized. Here, we identify a senescent subset within previously-defined cancer-associated fibroblasts (CAFs) in pancreatic ductal adenocarcinomas (PDAC) and in premalignant lesions in mice and humans. Senescent CAFs isolated from mouse and humans expressed elevated levels of immune-regulatory genes. Depletion of senescent CAFs, either genetically or using the Bcl-2 inhibitor ABT-199 (venetoclax), increased the proportion of activated CD8+ T cells in mouse pancreatic carcinomas, whereas induction of CAF senescence had the opposite effect. Combining ABT-199 with an immune checkpoint therapy regimen significantly reduced mouse tumor burden. These results indicate that senescent CAFs in PDAC stroma limit the numbers of activated cytotoxic CD8+ T cells, and suggest that their targeted elimination through senolytic treatment may enhance immunotherapy.

Cellular senescence serves as a central mechanism preventing tumorigenesis, by restricting the proliferation of damaged and oncogene-expressing cells[1–3]. However, senescent cells that persist in tissues during aging, and in the context of a variety of pathologies, exert complex non-cell-autonomous effects, which were shown to contribute to disease development[1–3]. Senescent cells are often detected within cancer lesions, both at premalignant stages and in advanced tumors, and their formation is thought to be due to stresses experienced by the cancer cells such as DNA damage and oncogene activity, as well as in response to cancer therapies[4–7]. It has become clear that the functions of senescent cells in cancer are complex, and

may take both tumor-promoting and tumor-suppressive forms. Furthermore, senescence can occur in tumor cells as well as in stromal cells, and in some cases is induced via signals between distinct cell types within lesions[6–8]. Uncovering the functions of senescent cells in different cancer settings is therefore of critical importance, and there is an ongoing effort to identify and develop senolytic drugs to eliminate senescent cells, or senoblockers, to inhibit their activity[3,9].

The effects of senescent cells on the tumor immune micro-environment are a topic of particular interest. Spontaneous and therapy-induced senescence of cancer cells has been shown to stimulate anti-tumor immune responses through several mechanisms,

including NK and T cell recruitment and activation, and elevated interferon pathway activity and antigen presentation[10–14]. However, in other settings, senescent cells have been shown to contribute to the generation of an inflammatory and immune-suppressed microenvironment through the senescence-associated secretory phenotype (SASP), thereby promoting tumorigenesis[15–18]. These diverse and seemingly contradictory effects of senescent cells illustrate the complexity of their functions in the tumor microenvironment (TME) and the need to dissect the roles of specific senescent cell subsets and their network of interactions.

Pancreatic ductal adenocarcinoma (PDAC) remains a major clinical challenge, with poor survival rates and few advances in therapy[19,20]. Immune checkpoint therapies (ICT) have not been successful in the treatment of this disease, to date. This is attributed to the immunosuppressive nature of the microenvironment in these tumors, which often carry a rich stroma characterized by low infiltration levels of CD8+ T cells and large numbers of myeloid-derived suppressor cells[19,20]. There is an urgent need to identify novel means to stimulate cytotoxic T cell infiltration and activation in this disease. Cancer-associated fibroblasts (CAFs) are a central component of the PDAC stroma, and there has been a major effort to characterize CAF subtypes and their roles, in this disease and in other cancer types[19–21]. While the main CAF types include inflammatory and myofibroblastic CAFs (iCAFs and myCAFs, respectively)[22], other categories, including antigen-presenting CAFs (apCAFs)[23] and populations marked by FAP, CD105, and LRRC15, have also been proposed to classify functional cell subsets[24–26]. There is a need to determine how different CAF subtypes modulate the TME, and whether they can be therapeutically targeted.

Senescent cells have been detected in PDAC, particularly in early disease stages, and in response to therapy[18,27–29]. Whether senescence of cells in the tumor stroma, and, specifically, of CAFs, contributes to PDAC development and progression is largely unknown.

Here, in analyzing senescent cells in human PDAC samples and in mouse models of the disease, we identified a substantial fraction of senescent cells in the tumor stroma, and found that it was comprised mostly of CAFs, representing different CAF subtypes. Dissecting the functions of the senescent CAFs, we found that their presence in the tumor stroma contributes to reduced numbers of activated cytotoxic CD8+ T cells, and that their elimination potentiates the response of tumors to ICT in a mouse model of the disease.

## Results

### Senescent CAFs are present in the stroma of developing PDACs

We set out to identify cells expressing senescence features in the course of pancreatic cancer development. Previous studies have identified senescent cells in Kras-driven mouse models of pancreatic cancer progression[18,27,28]. We expanded this work first by analyzing pancreatic sections from *Ptf1a-CreER; Kras*[lsl-G12D/+] mice, which express the oncogenic Kras[G12D] allele in acinar cells following CreER activation[30], and examining both epithelial and stromal lesion compartments. *Ptf1a-CreER; Kras*[lsl-G12D/+] mice develop acinar-to-ductal metaplasia and pancreatic intraepithelial neoplasia (PanIN) premalignant lesions within 3–6 months of Kras activation, while introduction of a conditional p53 null allele (*p53*[flox/+], together designated Kras+p53 hereafter) generates PDAC formation within 4–8 months of Kras activation (Fig. 1a). Staining of pancreatic sections for the central senescence activator and marker p16 (Cdkn2a) revealed the presence of p16+ cells within the epithelial and the stromal components of both PanINs and advanced PDACs (Fig. 1b, c and Supplementary Fig. 1a). The mean p16+ cell percentages were 17.6% and 19% in the epithelium and 8.7% and 12.5% in the stroma of PanIN and PDAC lesions, respectively, with substantial variability between tumors and regions (Fig. 1c). A similar pattern was observed upon staining of the same lesions for p21 (Cdkn1a), another Cdk inhibitor and marker of senescence (Supplementary Fig. 1b, c).

We found that the majority of stromal p16+ cells expressed the CAF marker Pdpn[23], indicating fibroblast identity, and that these cells were often located in proximity to the epithelial tumor lesions (Fig. 1d–f). While Pdpn marked the largest proportion of stromal CAFs (Supplementary Fig. 1d, e), p16 expression was detected in cells expressing markers that have been associated with different CAF subsets: fibroblasts expressing Pdgfra, which have been suggested in some settings to mark inflammatory CAFs (iCAFs), fibroblasts expressing α-SMA (Acta2), which mark myofibroblastic CAFs (myCAFs)[22,23], and cells expressing both markers (Fig. 1g, h). Whereas a fraction of p16-negative CAFs showed labeling for BrdU, marking proliferation, p16+ CAFs did not, further supporting the senescent state of the latter (Fig. 1i, j). Together these findings suggest that CAFs of different subtypes can enter a p16+, non-dividing, senescent state.

To better characterize the stromal senescent cell fraction, we conducted FACS analysis of dissociated mouse pancreata carrying Kras-driven lesions, using a fluorescent substrate for senescence-associated β-galactosidase (SA-βGal) activity, which marks senescent cells, combined with cell lineage markers. We found that SA-βGal+ cells were found in both the epithelial (EpCAM+) cell fraction and the non-epithelial (EpCAM-) (stromal) cell fraction (Fig. 1k, l and Supplementary Fig. 1f), whereas in control mice much fewer SA-βGal+ cells were detected (Fig. 1k). A mean of 15.5% of CAFs stained positive for SA-βGal (Fig. 1k, l). The sorted SA-βGal+ cell fraction of both the epithelial and the CAF populations showed elevated p16 expression when measured by qRT-PCR, supporting their senescent state (Fig. 1m). These analyses indicate that CAFs expressing central senescence markers are present within pancreatic cancer lesions, and that there is inter-tumoral and inter-regional variability in their numbers.

### Identification and isolation of senescent CAFs from human pancreatic cancers

We next tested whether senescent CAFs could also be detected in human pancreatic cancer. We first stained sections of human PDAC samples and pancreatic premalignant lesions for p16. Most premalignant lesions and cancers showed the presence of p16+ cells in the epithelial or stromal compartments (23 of 29 stained) (Fig. 2a, b, Supplementary Fig. 2a and Supplementary Table 1a). In PDAC, which often carry mutations in the gene encoding p16 (*CDKN2A*), a subset of samples (6 of 12) showed p16+ cells only in the stroma (Fig. 2a). Stromal p16+ cells showed fibroblastic morphology and expressed vimentin, indicating CAF identity (Fig. 2b, c).

To identify and isolate senescent CAFs directly from human patients, we dissociated live PDAC specimens following pancreatic surgery. All patients underwent pancreatectomy following previous chemotherapy or radiation treatment (Supplementary Table 1b). We sorted by FACS senescent and non-senescent PDAC CAFs after staining for SA-βGal together with the CAF marker CD90, and excluding epithelial and immune cells (EpCAM- CD45-) (Fig. 2d and Supplementary Fig. 2b). An average of 19.3% of CAFs were SA-βGal+ (range 9–32%), in five different patient samples (Fig. 2e). mRNA from matched SA-βGal+ and SA-βGal- CAF fractions was successfully isolated from four patients, and qRT-PCR confirmed the increased expression of p16 in the SA-βGal+ fraction, validating their senescent state (Fig. 2f).

We next studied a published scRNA-seq dataset of human PDAC[31], to test whether senescent cells could be identified within the CAF population. A total of 3492 CAFs from 21 PDAC patients formed a large cluster in which cell subsets expressing iCAF, myCAF and apCAF gene signatures[23], as well as other CAF markers, were observed (Fig. 2g, Supplementary Fig. 3a–c and Supplementary Table 2). To identify senescent CAFs, we used a modified version of a previously applied senescence gene signature that combines genes encoding several CDK inhibitors (including p16 and p21), and low levels of cell-cycle genes[32] (Supplementary Table 2). We detected CAFs positive for the senescence signature in all 21 subjects, with an average of 14.4% of cells

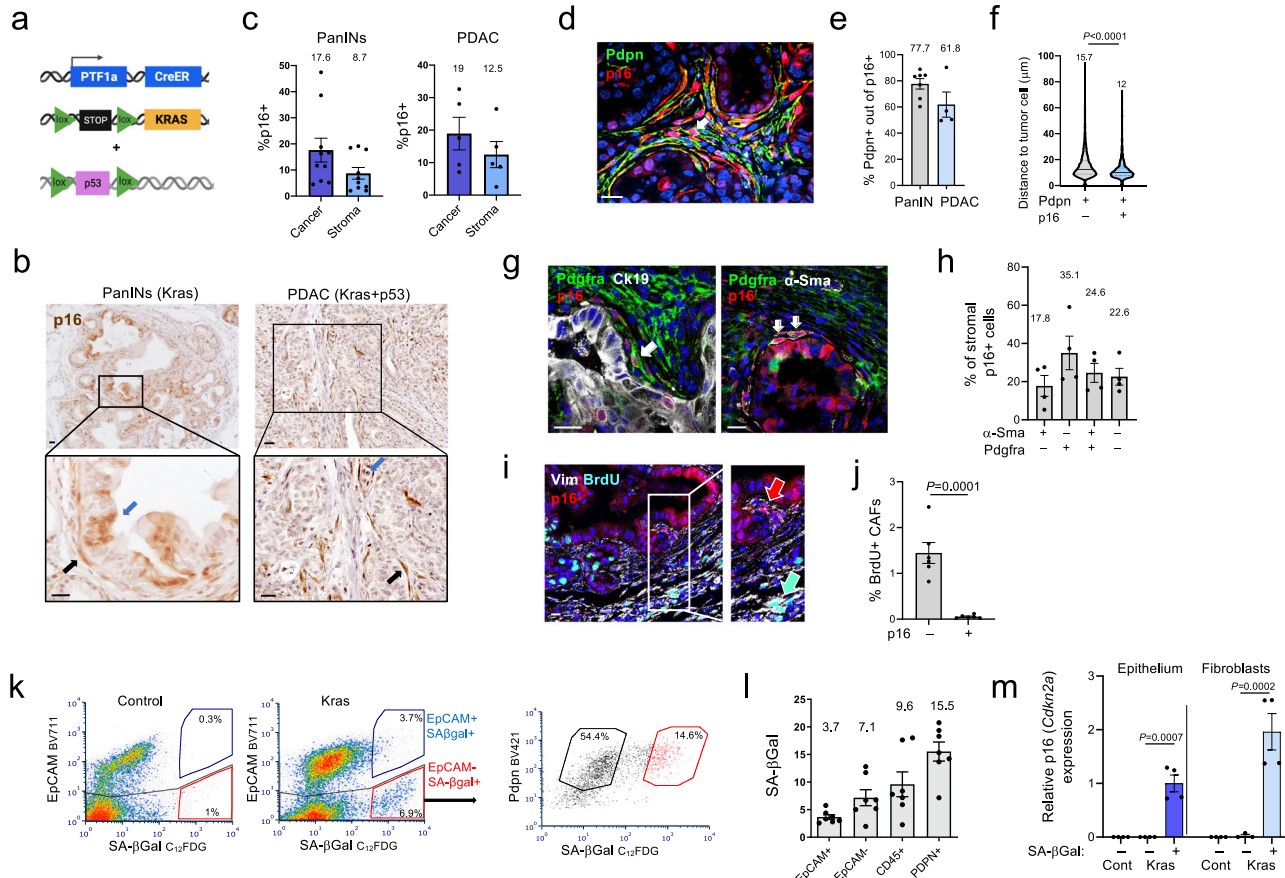

**Fig. 1 | Senescent CAFs are present in mouse pancreatic cancer lesions.**
**a** Diagram of mouse lines used for the generation of premalignant and malignant pancreatic carcinomas. **b** Stain for the senescence marker p16 in sections of PanIN lesions and PDAC from the indicated mice. Blue arrows indicate p16+ epithelial cells, black arrows indicate p16+ stromal CAFs. **c** Percentages of p16+ cells in epithelial (cancer) and stromal compartments of pancreatic lesions in Kras and Kras +p53 mice (PanINs and PDAC respectively), measured by image analysis. $n = 10$ PanIN, $n = 5$ PDAC individual lesions. **d** Co-stain of pancreatic lesions from Kras-activated mice for p16 and Pdpn, which marks stromal CAFs. **e** Percentages of Pdpn + cells out of stromal p16+ cells in PanIN ($n = 6$) and PDAC ($n = 4$) lesions. **f** Distances of p16-negative ($n = 6811$) and p16+ ($n = 898$) Pdpn+ CAFs to nearest cancer cell in three mouse tumors. Numbers indicate means, lines indicate median and quartiles. $t$ test. **g** Co-stains of pancreatic lesions from Kras-activated mice for p16, Pdgfra, α-Sma and CK19 marking epithelial cells, as indicated. Arrows indicate examples of p16+ Pdgfra+ CAFs and α-Sma+ CAFs. **h** Percentages of cells expressing Pdgfra, α-Sma, both, or neither, out of stromal p16+ cells in Kras-driven lesions. $n = 4$ tumors.

**i** Co-stain of pancreatic lesions for vimentin (Vim), marking fibroblasts, with p16 and BrdU, labeling proliferating cells. Cyan and red arrows indicate proliferating p16- CAFs and non-proliferating p16+ CAFs, respectively. **j** Percentage of BrdU+ cells among p16-negative and p16+ CAFs, identified by Pdpn stain. $n = 4$ tumors.
**k** FACS analysis of dissociated pancreatic lesions from Kras-activated mice. Left panels show stain for the senescence marker SA-βGal and the epithelial marker EpCAM, in control and in Kras-activated mice (Kras). Gates indicate %SA-βGal+ of EpCAM+ and EpCAM- fractions. Panel on right shows SA-βGal stain of Pdpn+ CAFs (EpCAM-, CD45-, CD31-). Values indicate percentage of SA-βGal- and SA-βGal+ cells out of Pdpn+ cells. **l** Percentages of SA-βGal+ cells within indicated cell fractions from Kras mouse pancreata, analyzed by FACS. $n = 7$ mice. **m** mRNA levels of p16 (Cdkn2a) measured by qRT-PCR in SA-βGal- and SA-βGal+ cell fractions of epithelial and fibroblast cells isolated from Kras-activated or control (Cont) mice.
$n = 4$ samples from two mice. $t$ test. Scale bars = 20 μm. In fluorescent images blue labels DNA. Graphs in (**c**, **e**, **h**, **j**, **l**, **m**) indicate mean ± SEM.

(range 3.2–31.7%), consistent with our stains and FACS analysis (Fig. 2h, i). Senescent CAFs were found among all three CAF subpopulations, representing 17.5%, 9.7%, and 5% in iCAFs, myCAFs, and apCAFs, respectively (Fig. 2h, j). Together these findings indicate that senescent stromal CAFs are present in a substantial portion of human pancreatic premalignant lesions and carcinomas, and represent different subtypes.

**Senescent CAFs express immune regulatory genes**
We next set out to identify the gene expression characteristics of senescent CAFs. We first conducted transcriptomic profiling of CAFs from Kras-driven mouse lesions. We FACS-isolated SA-βGal+ and SA-βGal- CAFs (Pdpn+, EpCAM-, CD45-, CD31-) from pancreata of mice in which Kras was activated, and compared their transcriptomes. The SA-βGal+ senescent CAFs showed reduced expression of cell-cycle genes and increased expression of ECM components and modifiers, of

cytokines associated with the SASP, of interferon-response genes and NFκB targets, and of genes associated with response to hypoxia and angiogenesis (Fig. 3a, b). A similar profile was obtained in senescent CAFs isolated from PDAC tumors that developed in Kras+p53 mice (Supplementary Fig. 4a, b).

We analyzed the transcriptomes of CAFs isolated from human patients (Fig. 2d–f). We found that the SA-βGal+ CAFs in these samples expressed elevated levels of stress-response and angiogenesis genes sets, and of immune regulatory gene sets, including IL6, IL17, TGFβ and AP1 pathways, as well as of negative regulators of proliferation (Fig. 3c), findings similar to those observed in the mouse CAFs.

We also developed cell-culture models for controlled induction of senescence in primary mouse and human PDAC CAFs. Primary CAFs were isolated from mouse PDAC tumors or from human PDAC patients, and were infected with a lentivirus carrying the CreER inducible recombinase, together with a virus carrying the immortalizing

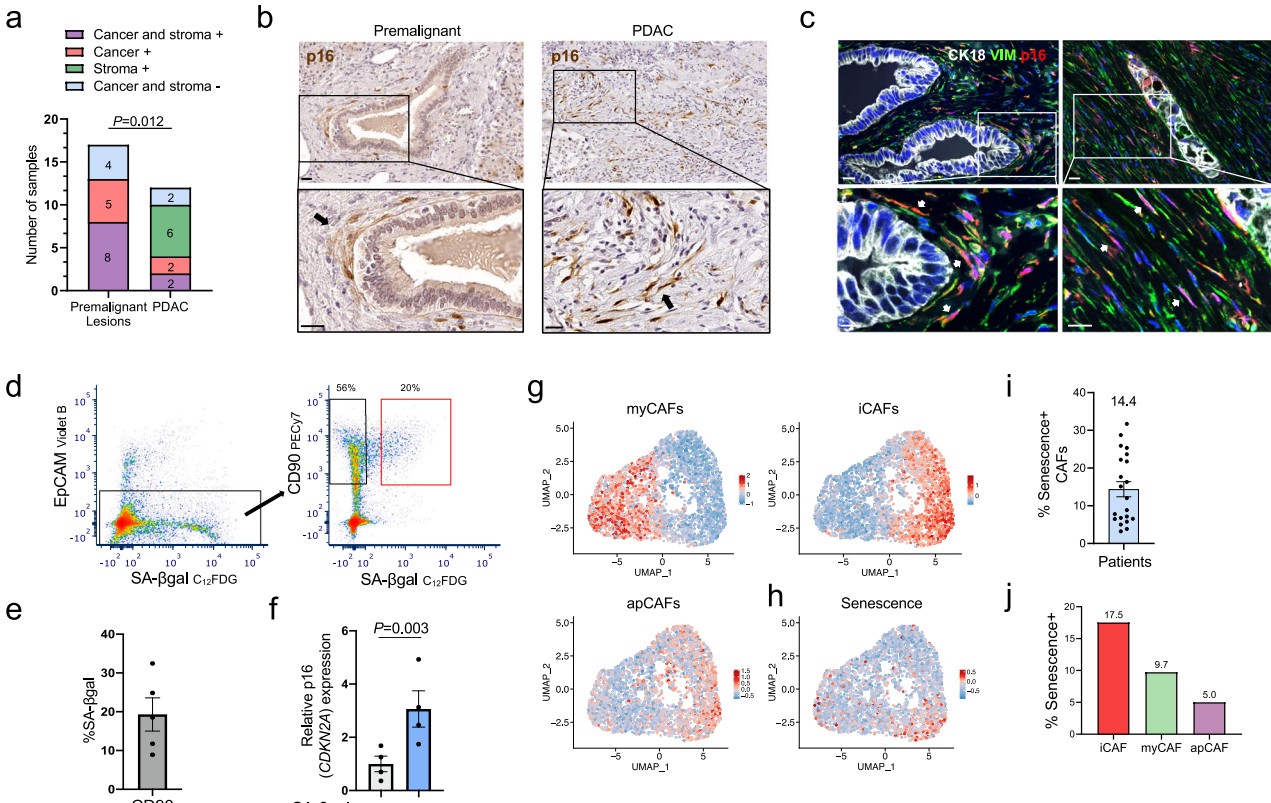

**Fig. 2 | Senescent CAFs are present in human pancreatic cancer lesions.**
**a** Numbers of human premalignant lesions and PDACs in which p16+ cells were detected in the tumor or stromal compartments, in both, or in neither, as indicated. *P* value, Chi-squared, refers to the difference in distribution between the lesion types. **b** Sections of human premalignant (low grade PanIN) (left) and PDAC (right) samples, stained for p16, as scored in panel a. Black arrows indicate p16+ CAFs. **c** Co-stain of human PDAC sections for p16, the epithelial marker CK18, and vimentin (VIM), as scored in (**a**). Arrows indicate p16+ CAFs. **d** FACS isolation of SA-βGal- and SA-βGal+ CAFs from a representative human PDAC sample. CD90 marks fibroblasts. Plot on right is gated for EpCAM-CD45- cells. Gate values are calculated out of CD90+ fraction. **e** Average percentage of SA-βGal+ cells in CAFs from different patients. Mean of *n* = 5 tumors ± SEM. **f** Levels of p16 (*CDKN2A*) mRNA in matched SA-βGal- and SA-βGal+ CAFs isolated from human PDAC patients. Mean of *n* = 4 tumors ± SEM. *t* test. **g** UMAPs showing a scRNA-seq cluster of 3492 individual PDAC CAFs from 21 patients, obtained from ref. 31. Red color indicates the relative scores of iCAF, myCAF and apCAF expression signatures in individual CAFs. **h** UMAP of same CAFs, indicating relative expression score of the senescence signature. **i** Percentage of CAFs scoring positive for the senescence signature in each of the individual subjects included in the scRNA-Seq analysis. Mean of *n* = 21 patients ± SEM. **j** Percentages of CAFs classified to the indicated subtypes which score positive for the senescence signature. Scale bars = 20 μm.

gene SV40 large-T-antigen (which inhibits p53 and Rb, hereafter LT), and a lentivirus carrying the telomerase catalytic subunit hTERT (introduced only into human CAFs), both flanked by loxP sites[33] (Supplementary Fig. 4c). Treatment of the cells with 4-OH-tamoxifen (TAM) led to CreER activation and excision the of the LT and hTERT cassettes, inducing the treated CAFs to undergo senescence (Fig. 3d, e and Supplementary Fig. 4c–e). We then conducted transcriptome analysis of non-senescent and senescent CAFs.

We cross-compared the gene sets enriched in the transcriptomes of the different sets of senescent CAFs described above, and found that the main gene sets were conserved across models and species. These included upregulation of genes in the p53-response pathway, UV, hypoxia and wound-healing responses, and genes in immune regulatory pathways: TNFα signaling through NFκB, IL6-STAT3, interferon, complement, inflammation and TGF-β (Fig. 3f). qRT-PCR analysis validated the elevation of mRNAs encoding various cytokines and matrix modifiers in the mouse and human cultured PDAC CAFs (Supplementary Fig. 4f,g). These included genes encoding SASP factors such as IL1α, IL1β, IL6, CSF1, CXCL2, CXCL10, PD-L1, complement factor C3, matrix modifiers TIMP1,2 and SULF2, and MHC class I molecules (HLA-I, B2M). Stimulation of the CAFs with IFNγ led to increased levels of most cytokine-encoding and MHC-related genes, with the senescent

CAFs showing further elevation, consistent with heightened interferon responsiveness[13,14] (Supplementary Fig. 4f, g). Mass spectrometry analysis of conditioned medium from senescent and non-senescent human CAFs supported the increased secretion of these cytokines and matrix modifiers by the senescent CAFs (Supplementary Fig. 4h).

Analysis of the scRNA-seq dataset obtained from PDAC patients provided further support for these findings. CAFs with a positive senescence score (Fig. 2h, i) expressed elevated levels of gene sets enriched for immune regulatory pathways including AP1 target genes, IL18, IL4, and IL13 pathways, as well as hypoxia, angiogenesis, and p53 response gene sets, demonstrating a strong correspondence to the gene expression profiles we obtained experimentally (Fig. 3g).

Together these results establish that senescent CAFs in PDAC express elevated levels of genes that modulate the tumor immune microenvironment, as well as genes involved in ECM organization, stress-responses, and vasculature growth.

## CAF senescence negatively correlates with T cell content in mouse and human tumors

To test whether there was a link between the presence of senescent CAFs and T cell infiltration – a central feature of the immune TME – we analyzed the numbers of stromal p16+ CAFs in a panel of mouse PDAC

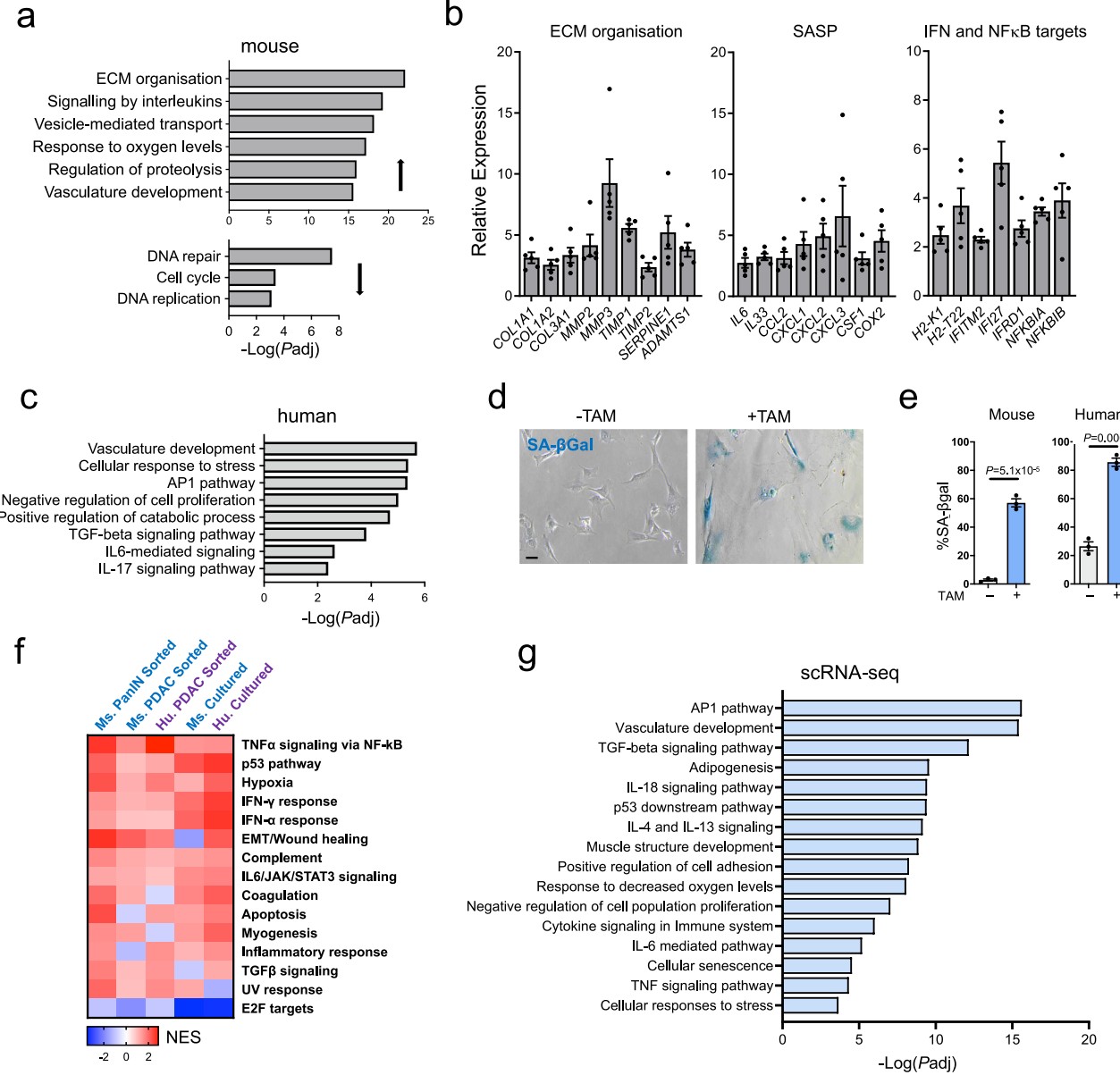

**Fig. 3 | Senescent CAFs express elevated levels of immune regulatory genes.**
**a** Gene sets up- and down-regulated in SA-βGal+ CAFs isolated from Kras-activated mouse pancreata, compared to matched SA-βGal- CAFs, as measured by mRNA-seq. x axis indicates -$\log_{10}$ $P$adj values, calculated by Metascape with Bonferroni method multiple hypothesis correction. **b** Selected upregulated genes in the same SA-βGal+ CAFs. y axis indicates fold change relative to SA-βGal- CAFs, as measured by mRNA-seq. Mean of $n = 5$ tumors ± SEM, differences for all are under a threshold of $P$adj < 0.1 calculated by DESeq2 (Benjamini-Hochberg FDR correction). **c** Gene sets upregulated in SA-βGal+ CAFs isolated from human subjects, relative to matched SA-βGal- CAFs. x axis indicates −$\log_{10}$ $P$adj values, calculated by Metascape with Bonferroni method multiple hypothesis correction. **d** SA-βGal staining marking senescence of primary cultured human PDAC CAFs following activation of CreER by 4-OHT (TAM) to eliminate hTERT and LT expression. Scale bar = 20 μm.

**e** Percentage of SA-βGal+ cells in cultured mouse and human CAFs treated or untreated with TAM. Mean of $n = 3$ replicates ± SEM. $t$ test. **f** Gene sets enriched in five analyzed datasets comparing senescent to non-senescent CAFs: cells isolated directly from Kras-induced mice, from Kras+p53 mice, or from human patients, and mouse and human CAFs induced to undergo senescence in culture. Colors indicate normalized enrichment scores calculated by GSEA, with positive values indicating increased expression in senescent CAFs. Shown are sets with $P$adj < 0.25 (Benjamini-Hochberg FDR correction) in at least 3 out of the 5 datasets. NES normalized enrichment score. **g** Gene sets upregulated in CAFs within the scRNA-seq dataset which score positive for the senescence signature. x axis indicates −$\log_{10}$ $P$adj values, calculated by Metascape with Bonferroni method multiple hypothesis correction.

cell lines derived from the *PDX1-Cre; Kras*[G12D/+]*; p53*[R172H/+] model (known as KPC cells)[23,34]. These lines offer the opportunity to focus on stromal senescence, since the tumor cells, which lack p53 activity and often lose p16 expression[34], are less susceptible to spontaneous senescence. Previous work has characterized a panel of KPC lines, which were shown to exhibit variable levels of T cell infiltration and activity[34]. p16 expression was detected in the stroma of tumors derived from various lines, when transplanted in the mouse pancreas. Similar to the transgenic model, the majority of p16+ stromal cells were Pdpn+ CAFs

(70%), with the remainder representing endothelial cells (9%) and immune cells (8%), approximately half of which were macrophages (Fig. 4a, b). Interestingly, we noted that tumors with high percentages of p16+ CAFs (Pdpn+) showed low levels of CD3+ T cell infiltration, and vice versa (Fig. 4c, d). In contrast, the percentage of all CAFs out of the stroma did not correlate well with T cell numbers (Fig. 4e), supporting the specific link to CAF senescence. A similar trend was observed relative to CD8+ T cell numbers, although this was not statistically significant (Supplementary Fig. 5a). A negative correlation between the

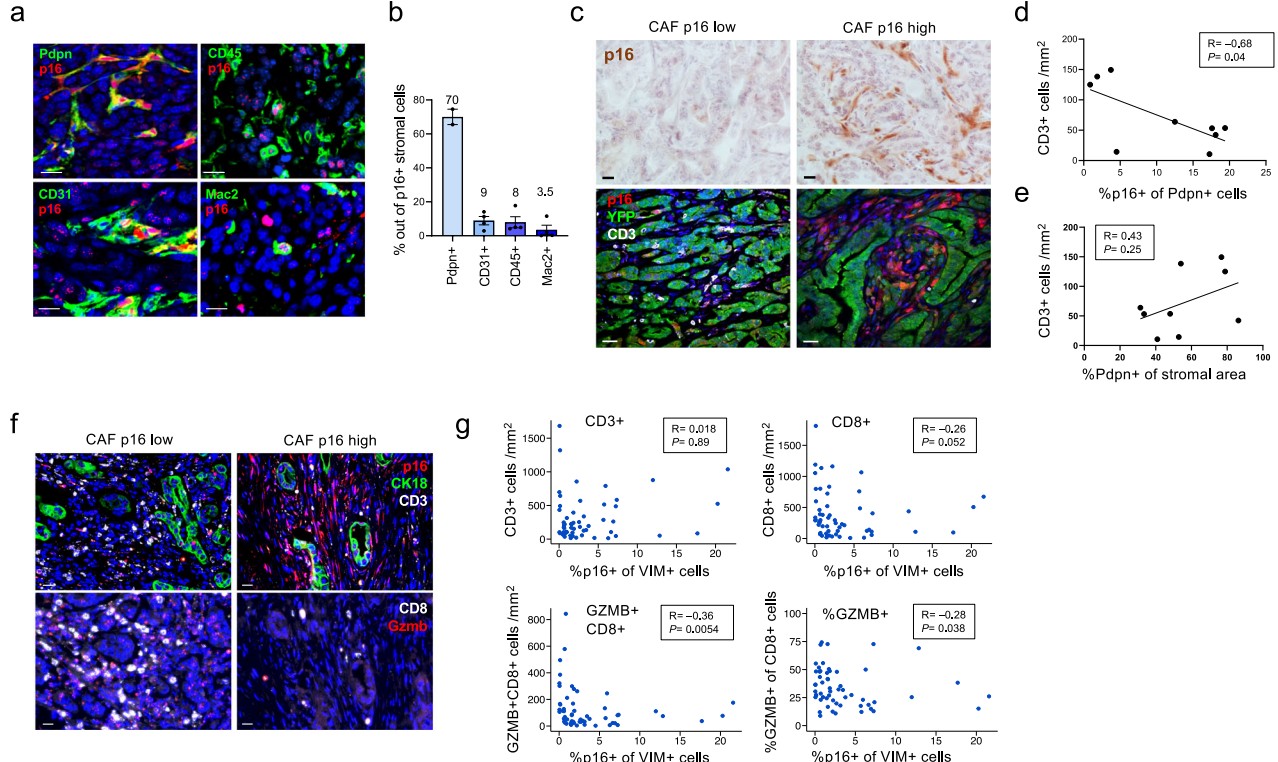

**Fig. 4 | CAF senescence correlates with reduced T cell presence in mouse and human PDACs. a** Co-staining of 6422c1 KPC xenograft tumors for p16 together with either Pdpn, marking CAFs, CD31 marking endothelial cells, CD45 marking immune cells, and Mac2, marking macrophages. Images show examples of double-positive cells of each type. **b** Percentages of cells expressing the indicated lineage markers out of p16+ stromal cells in the 6422c1 KPC xenografts, as shown in panel a. Mean of *n* = 2–4 tumors as indicated ± SEM. **c** Sections of tumors that developed from different KPC PDAC lines that were injected into mouse pancreata, stained for p16 (top), or co-stained for p16, CD3, marking T cells, and YFP, marking the cancer cells (bottom). Tumors on left, which developed from the 6555c3 line, show low levels of p16+ CAFs and high T cell numbers, whereas tumors on right, which developed from the 6422c1 and 6499c3 lines, show high levels of p16+ CAFs and low T cell numbers. **d** Correlation between percentage of p16+ cells out of CAFs, and CD3+ T cell numbers per mm², in individual tumors grown from distinct KPC

lines, quantified by image analysis. R value indicates Pearson correlation with calculated *P* value, with line indicating linear regression. *n* = 9 tumors. **e** Correlation between percentage of Pdpn+ CAFs out of stromal area, and CD3+ T cell numbers per mm² tumor, in same tumors as in (**d**). *n* = 9 tumors. **f** Images of regions in human PDACs co-stained for p16, CD3, and CK18 marking tumor cells (top) or for CD8 and GZMB (bottom, same regions in sequential sections). Left images show region with low p16+ CAF content and high CD3+, CD8+ and GZMB+ CD8+ cell numbers; right images show region with high p16+ CAF content and low T cell numbers. Quantifications are provided in (**g**). **g** Correlation between percentages of p16+ cells out of VIM+ CAFs in 57 tumor regions in 11 human PDACs, and T cell numbers in same regions, scored by image analysis. y axes in graphs indicate numbers CD3+, CD8+ or GZMB+ CD8+ T cells per mm² as indicated; bottom right graph shows percentage of GZMB+ out of CD8+ T cells in y axis. R values indicate Spearman correlation with calculated *P* value. Scale bars = 20 µm.

presence of p16+ CAFs and infiltrating T cells was also observed between regions within the same tumors, in the Kras-driven transgenic model (Supplementary Fig. 5b, c). These correlations suggested an interaction between senescent CAFs and T cells in the TME.

We next examined whether the rates of regional CAF senescence correlated with T cell infiltration in human PDAC. The stromal microenvironment in PDAC is highly heterogeneous between tumors and different regions in the TME[35]. We scored 57 regions in 11 human PDAC samples for the percentage of CAFs that were p16+, and for the numbers of CD3+ and CD8+ T cells in each. While total CD3+ T cell numbers did not show a significant correlation with p16+ CAF rates, CD8+ T cell numbers showed a negative correlation with senescent CAF percentages (Fig. 4f, g). Furthermore, the numbers of granzyme B-expressing (GZMB+) cells among CD8+ cells, representing activated cytotoxic (effector) CD8+ T cells[36–38], showed a more pronounced negative correlation with CAF senescence rates, as did the percentage of GZMB+ out of CD8+ T cells in these regions (Fig. 4f, g). This indicated that in human PDAC, p16+ CAF rates are associated more specifically with reduced presence of activated CD8+ T cells. Overall, these correlations raised the question of whether there is a functional relationship between CAF senescence and T cell numbers and function in the tumors.

## Induction of senescence in PDAC CAFs in vivo reduces numbers of activated CD8+ T cells

To test the potential causative effects of CAF senescence on T cell infiltration and activation, we used the mouse CAFs carrying the inducible senescence gene system described above. We co-transplanted KPC tumor cells into wt mouse pancreata together with mCAFs carrying the inducible senescence cassette, and labeled with mScarlet. We used a KPC line (6555c3) that forms tumors with high rates of T cell infiltration and low stromal senescence. We treated the mice with tamoxifen one and three days after transplantation, to activate CreER and senescence of the CAFs (Fig. 5a). One week after transplantation, pancreata were resected and regions containing the forming tumor were identified. The injected mScarlet CAFs were detected in the lesions, intermixed with the YFP-labeled tumor cells (Fig. 5b). Co-staining analyses indicated that in the tamoxifen-treated mice the mScarlet+ CAFs showed loss of expression of Large T-antigen, indicating successful recombination, together with increased p21 expression, and decreased proliferation (Supplementary Fig. 6a, b). This indicated that the senescence system was activated in the injected CAFs, yet likely not to the full extent observed in culture.

We next analyzed the immune composition of the forming tumors to assess whether senescence activation influenced tumor immune

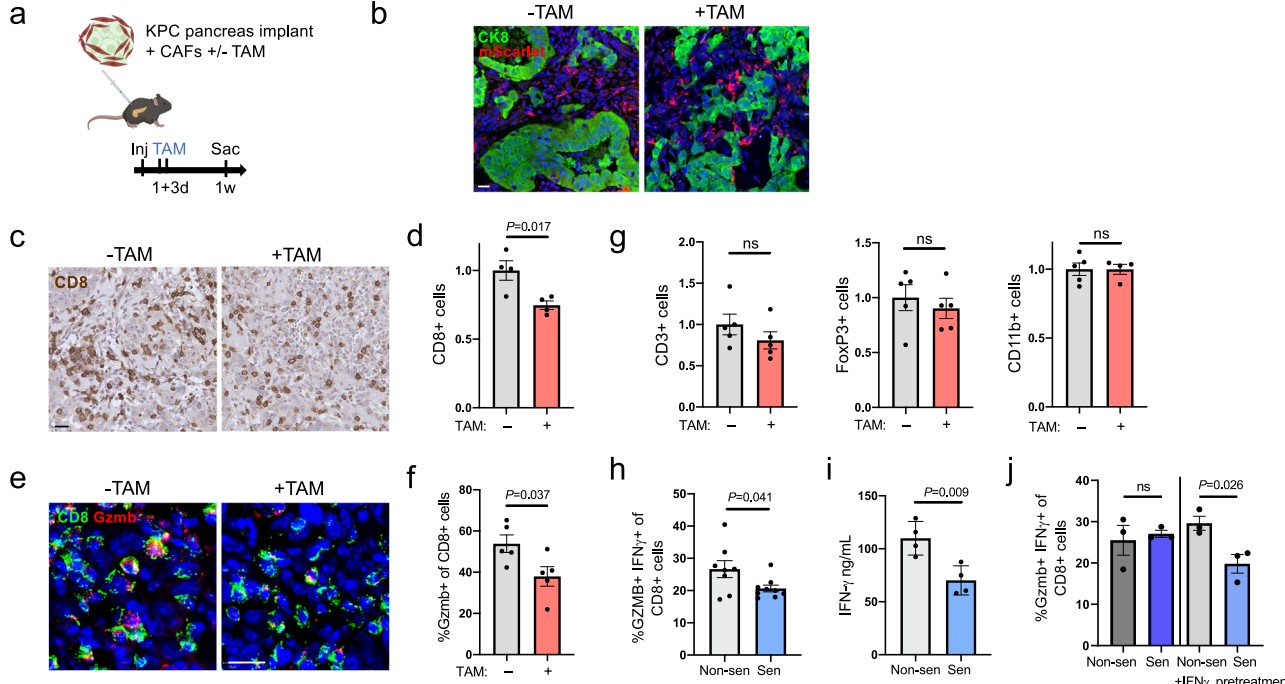

**Fig. 5 | Induction of CAF senescence reduces rates of activated CD8+ T cells.** **a** Diagram of experimental procedure: mScarlet-expressing mouse CAFs were co-injected into mouse pancreas with 6555c3 KPC tumor cells. Senescence was induced in CAFs in half of the mice through tamoxifen (TAM) treatment one and three days later, and tissues were collected one week after cell transplantation. **b** Lesion sections from untreated (-TAM) and tamoxifen-treated (+TAM) mice, further analyzed in panels c-g, 1 week after co-injection, stained for mScarlet marking the injected CAFs, and for CK8 marking the co-injected KPC cells. **c** Lesion sections from control and tamoxifen-treated mice stained for CD8. **d** Quantification of CD8+ T cells in lesions from untreated and tamoxifen treated mice. Values were measured by image analysis and are presented relative to mean of control mice. *n* = 4 tumors. **e** Lesion sections from untreated and tamoxifen-treated mice co-stained for CD8 (green) and Gzmb (red). **f** Percentages of Gzmb+ cells out of CD8+ T cells in lesions from untreated and tamoxifen-treated mice, scored by

image analysis. *n* = 5 tumors. **g** Quantification of total T cells (CD3+), regulatory T cells (FoxP3+), and monocytes (CD11b+) in same lesions. Values were measured by image analysis and are presented relative to mean of control mice. *n* = 5 tumors. **h** Percentage of IFNγ+ and GZMB+ human CD8+ T cells following treatment with conditioned media from non-senescent (Non-sen) or senescent (Sen) human PDAC CAFs, measured by FACS. *n* = 8 replicates from two experiments. **i** IFNγ levels secreted by human T cells following treatment with conditioned media from non-senescent or senescent human CAFs, measured by ELISA. *n* = 4 replicates. **j** Percentage IFNγ+ Gzmb+ mouse CD8+ T cells following treatment with conditioned media from non-senescent or senescent mouse CAFs, measured by FACS. Where indicated, CAFs were pretreated with IFNγ for 48 h prior to conditioned media collection. *n* = 3 replicates. All values represent mean ± SEM. Pairwise two-tailed *t* tests compare senescence to non-senescence in all graphs. Scale bars = 20 μm.

composition. We noted a decrease in the numbers of CD8+ T cells in the tumors from tamoxifen-treated mice, indicating that the senescent CAFs restricted CD8+ T cell numbers (Fig. 5c, d). Furthermore, the percentage of Gzmb+ cells among CD8+ cells was reduced in mice in which CAF senescence was induced (Fig. 5e, f). The total numbers of CD3+ T cells, CD11b+ monocytes and of FoxP3+ regulatory T cells were not significantly affected by senescence activation (Fig. 5g). These findings support the notion that CAF senescence limits the presence of activated CD8+ T cells in PDAC. In tumors grown for longer periods we found that the injected CAFs, due to their lower proliferation rates, were detected in low numbers or not at all, precluding the testing of the effects of their senescence.

We next tested whether senescent PDAC CAFs could directly influence T cell activation state, in an isolated culture setting. We treated human T cells with conditioned media (CM) from the non-senescent or senescent human PDAC CAFs described above, and measured T cell activation state. CD8+ T cells showed reduced activation rates when treated with the CM of senescent CAFs, as measured by IFNγ+ GZMB+ percentages scored by FACS, and IFNγ secretion levels measured by ELISA (Fig. 5h, i). Similarly, mouse CD8+ T cells treated with CM from senescent mouse PDAC CAFs also showed reduced activation compared to T cells treated with CM from non-senescent CAFs, yet this occurred only after prior treatment of the CAFs with IFNγ to stimulate their secretome (Fig. 5j). These results indicate that senescent CAFs can repress CD8+ T cell activity through

direct paracrine signaling, yet suggest that the senescent CAFs may respond to extrinsic immune-provided signals such as IFNγ to upregulate their secretome and, in turn, counteract CD8+ T cell activation.

We also tested whether senescent PDAC CAFs influence carcinoma cells directly. Several studies have demonstrated that senescent fibroblasts can stimulate carcinoma cell proliferation through SASP cytokines[8]. We therefore treated human patient-derived PDAC cells (line X252) with CM from non-senescent and senescent human CAFs. However, no significant effect on tumor cell proliferation was observed (Supplementary Fig. 6c). Similar results were observed in tumorspheres comprised of mouse KPC tumor cells mixed with senescent or non-senescent mouse PDAC CAFs (Supplementary Fig. 6d). These results do not support a model in which senescent CAFs directly stimulate PDAC cell proliferation.

### Elimination of senescent PDAC CAFs using the *Ink-ATTAC* model leads to increased numbers of activated CD8+ T cells

We next turned to study the effects of the removal of endogenously-formed senescent CAFs from growing tumors, using a targeted system for their elimination. We injected KPC tumor cells that induce high levels of stromal senescence (line 6422c1) into the pancreata of C57Bl6 *Ink-ATTAC* mice[39]. In these transgenic mice, an inducible apoptosis-inducing cassette is placed under the control of the p16 promoter, allowing specific elimination of p16+ cells through treatment with the AP20187 dimerizer (AP hereafter). Because the injected tumor cells do not carry

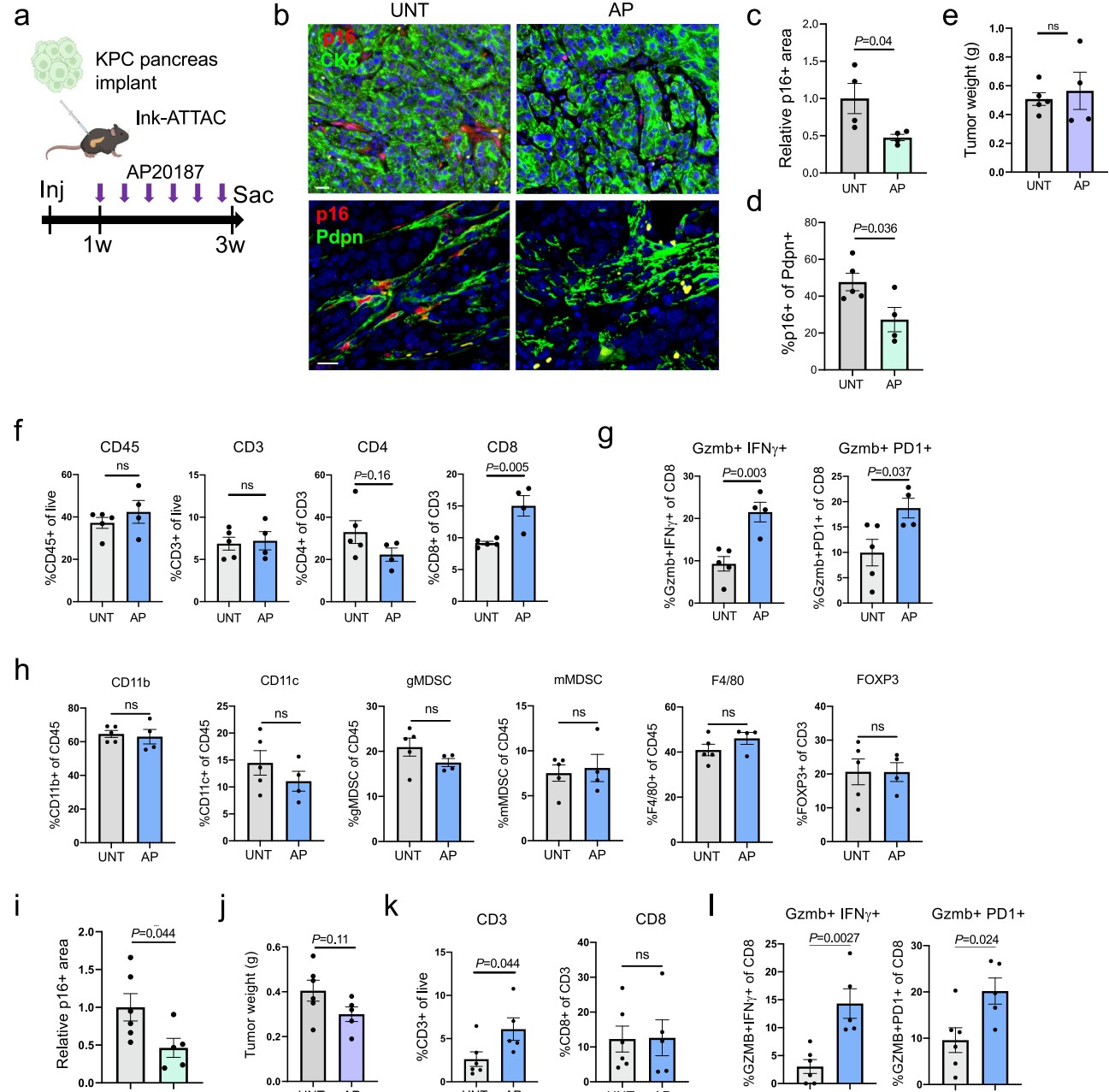

**Fig. 6 | Genetic elimination of senescent CAFs in transplanted PDACs leads to increased rates of activated CD8+ T cells. a** Diagram of experimental design. 6422c1 KPC cells, which generate tumors with high endogenous p16+ CAF content, were injected into the pancreata of *Ink-ATTAC* mice. Mice were treated with AP20187 three times a week during tumor growth. **b** Sections of tumors formed by KPC cells in mice either untreated (UNT) or treated with AP, co-stained for p16 and with CK8 (top), marking epithelial cells or with Pdpn (bottom), marking CAFs. **c** Relative stromal p16+ area in tumors formed by the KPC cells, with or without AP treatment. Values were measured by image analysis and normalized to mean of controls. $n = 4$ tumors, >1 mm²/tumor. **d** Percentage of p16+ cells out of Pdpn+ CAFs in same samples, scored by image analysis. $n = 5,4$ tumors. **e** Tumor weights upon sacrifice, in mice treated with AP or untreated. $n = 5,4$ tumors. **f** Quantification of indicated immune cell populations in tumors formed in control and AP-treated

mice, scored by FACS. $n = 5,4$ tumors. **g** Percentages of activated CD8+ T cells in same mice, scored with indicated markers by FACS. **h** Quantification of additional indicated immune cell populations in tumors formed in control and AP-treated mice, scored by FACS. $n = 5,4$ tumors. CD11b and CD11c mark monocytic populations, gMDSC – granulocytic myeloid-derived suppressor cells (Ly6G⁺Ly6C^low), mMDSC – monocytic myeloid-derived suppressor cells (Ly6G⁻Ly6C^high), F4/80 marks macrophages, FoxP3 marks regulatory T cells. **i** Relative stromal p16+ area in tumors formed by FC1199 KPC cells, with or without AP treatment, measured by image analysis. $n = 6,5$ tumors, >1 mm²/tumor. **j** Tumor weights upon sacrifice, in mice transplanted with FC1199 KPC cells treated with AP or untreated. $n = 6,5$ tumors. **k, l** Quantification of indicated T cell populations in FC1199 tumors formed in control and AP-treated mice, scored by FACS. $n = 6,5$ tumors. All values indicate mean ± SEM. *t* test. Scale bars = 20 μm.

the cassette, senescent cell elimination is restricted to endogenously formed senescent cells in the mouse, including the tumor stroma. Mice were treated with AP three times a week, during the last two weeks of three weeks of tumor growth (Fig. 6a). AP treatment reduced the numbers of p16+ stromal cells in tumors by approximately 50% (Fig. 6b, c). Since, as shown above, the large majority of senescent

stromal cells are fibroblasts, a corresponding decrease was observed in the percentage of p16+ Pdpn+ CAFs (Fig. 6b, d). Tumors formed in the mice treated with AP did not show a change in final mass (Fig. 6e). However, FACS analysis revealed that in AP-treated mice there was an increase in CD8+ T cell numbers in the tumors (Fig. 6f). Most strikingly, the percentage of activated cytotoxic CD8+ T cells was increased in the

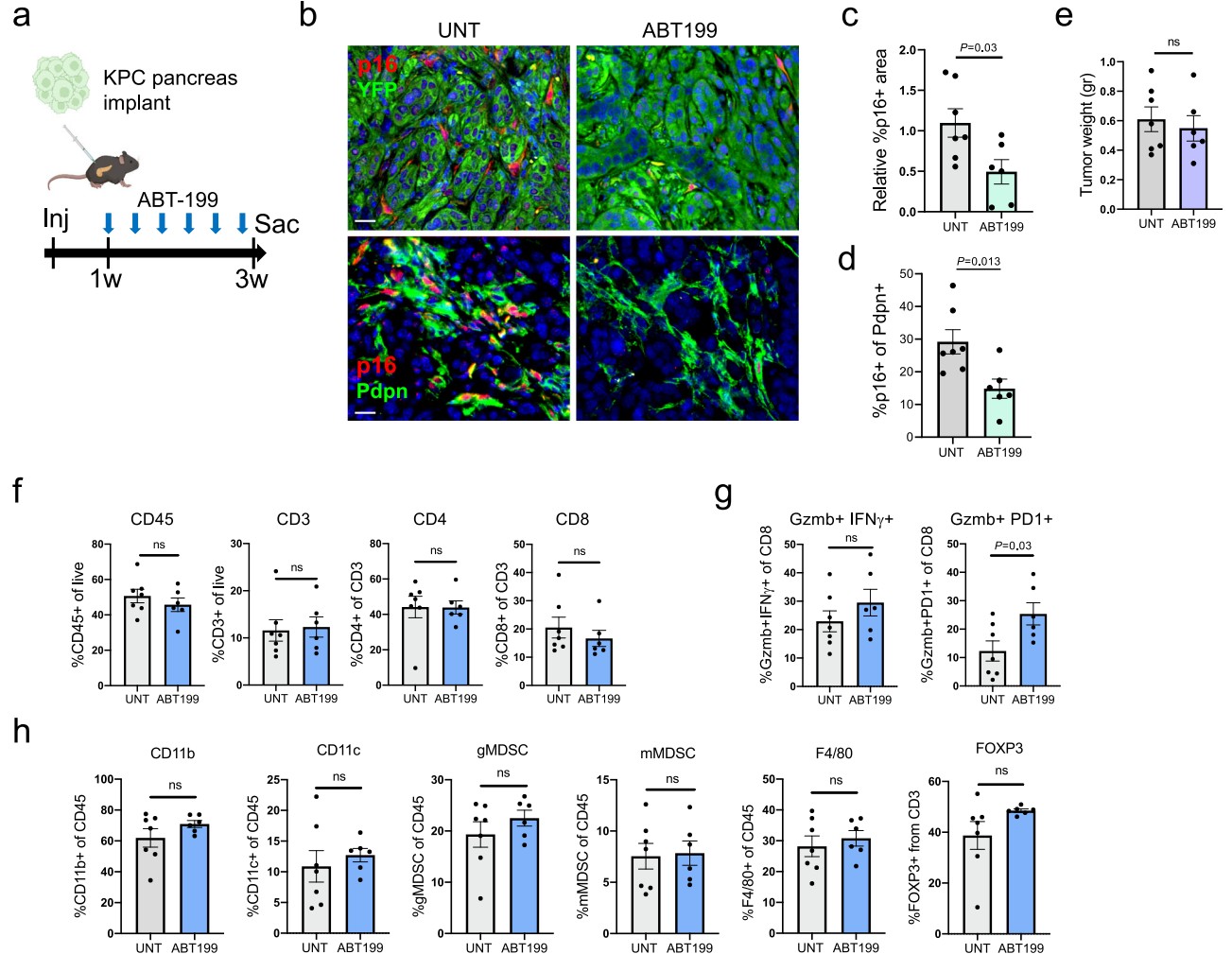

**Fig. 7 | Elimination of senescent CAFs in transplanted PDACs using ABT-199 increases activated CD8+ T cell rates. a** Diagram of experimental design. 6422c1 KPC cells were injected into wt mouse pancreata. Mice were treated with ABT-199 three times a week during tumor growth. **b** Sections of tumors formed by KPC cells in mice either untreated or treated with ABT-199, co-stained for p16 and for YFP (top), marking epithelial cells, or Pdpn (bottom) marking CAFs. **c** Relative stromal p16+ area in tumors with or without ABT-199 treatment. Values were measured by image analysis and normalized to mean of controls. $n = 7$ UNT, $n = 6$ ABT-199, >1 mm²/tumor. **d** Percentage of p16+ cells out of Pdpn+ CAFs in same samples, scored by image analysis. **e** Tumor weights upon sacrifice, in mice treated with ABT-199 or untreated. **f** Quantification of indicated immune cell populations in tumors formed in untreated and in ABT-199-treated mice, scored by FACS. **g** Percentages of activated CD8+ T cells in same mice, scored with indicated markers by FACS. **h** Quantification of indicated additional immune cell populations in tumors formed in untreated and in ABT-199-treated mice, scored by FACS. $n = 7,6$ tumors in (**c**–**h**). All values indicate mean ± SEM. $t$ test. Scale bars = 20 μm.

AP-treated tumors (Fig. 6g). Other immune cell types scored, including myeloid-derived suppressor cells (MDSCs), did not show significant changes (Fig. 6h and Supplementary Fig. 7). We observed similar results using another KPC line with high stromal p16 content (line FC1199) (Fig. 6i–l). Consistent with the *Ink-ATTAC* system eliciting non-inflammatory apoptosis, we did not observe evidence of overall tumor inflammation induced by the targeted cell killing (Supplementary Fig. 8). These findings indicate that senescent CAFs act to limit the numbers of activated cytotoxic CD8+ T cells in the growing tumors.

**Senolytic elimination of senescent PDAC CAFs with the Bcl2 inhibitor ABT-199 leads to increased numbers of activated CD8+ T cells**

We next set out to test whether senolytic treatment can achieve senescent CAF elimination and influence CD8+ T cell activation. Inhibitors of the Bcl2 anti-apoptotic protein family have been shown to be among the most effective senolytic drugs in mouse models[3,9]. We compared the expression of the four Bcl2-family proteins in Kras-driven lesions. We found that among these, Bcl2 itself was the most prominently expressed in stromal cells, whereas other

members – Bclxl, Bclw and Mcl1 – were expressed in subsets of epithelial tumor cells (Supplementary Fig. 9a). Consistent with this, ABT-737, which inhibits Bcl2, Bcl-xl and Bcl-w, reduced the numbers of p16+ cancer cells (as we have previously shown[18]), whereas the Bcl2-specific inhibitor ABT-199 achieved better elimination of the stromal senescent cells (Supplementary Fig. 9b, c). ABT-199, also known as venetoclax[40], is currently used for the treatment of hematopoietic malignancies including CLL and AML. Based on this finding we therefore proceeded to test the senolytic effects of ABT-199 treatment on the KPC xenograft model.

We implanted 6422c1 KPC tumor cells in mouse pancreata, and treated the animals with ABT-199 three times a week in the last two weeks of three weeks of tumor growth (Fig. 7a). Excised tumors showed a reduction of ~50% in the numbers of p16+ CAFs relative to untreated mice (Fig. 7b–d), similar to the efficiency of the *Ink-ATTAC* model, and this reduction did not have a significant effect on final tumor weights (Fig. 7e). Analysis of immune cell content revealed that while T cell numbers within the tumors did not substantially change, the percentage of activated CD8+ T cells was increased in the mice treated with ABT-199 (Fig. 7f, g). This effect was similar to that

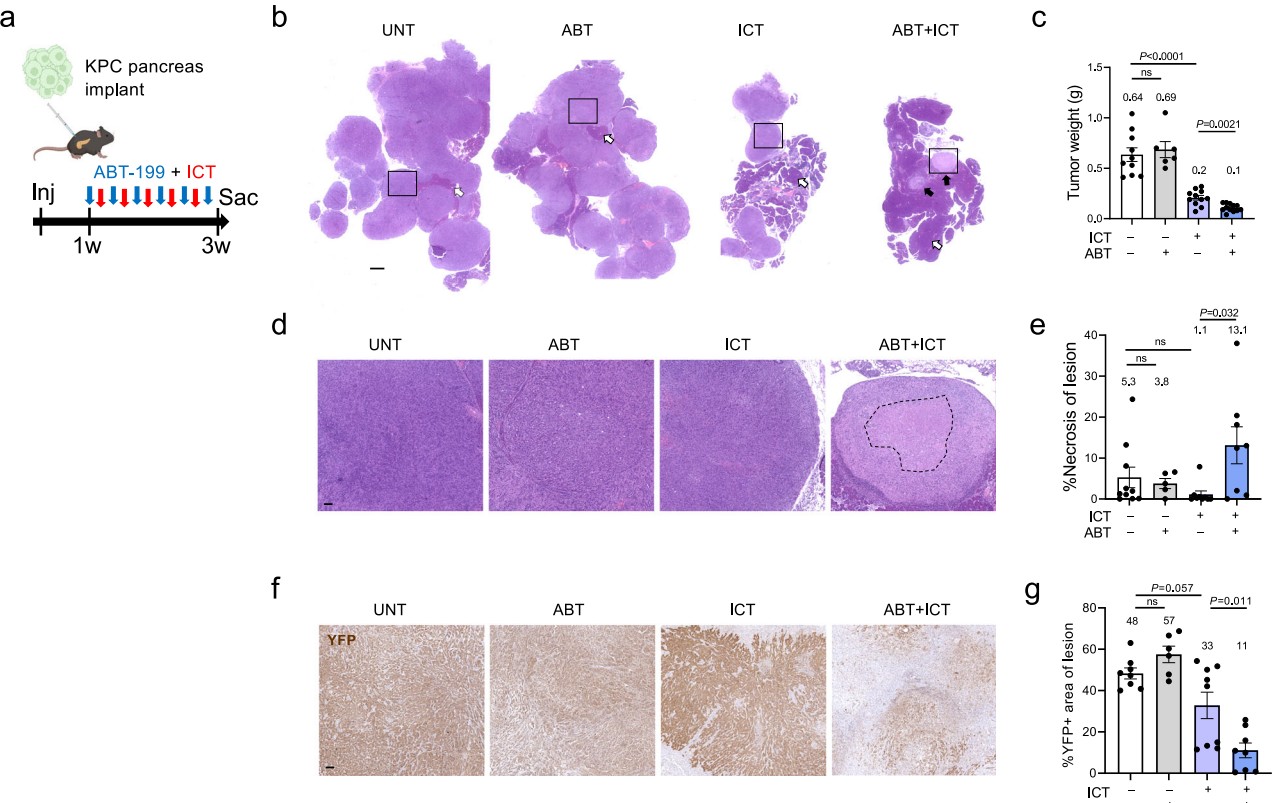

**Fig. 8 | Senolytic treatment enhances the response to immune checkpoint therapy. a** Diagram of experimental design. 6422c1 KPC tumor cells were injected into mouse pancreata. Mice were treated with ABT-199, with ICT including αCTLA4, αPD1 and αCD40, or with both treatment types. **b** H&E-stained images of representative pancreatic tissues in mice treated with the different protocols. Dark regions represent remaining normal acinar tissue (white arrows). Tumor lesions in ABT + ICT mice are indicated by black arrows. Scale bar = 1 mm. **c** Pancreatic tissue weights upon mouse sacrifice in the different treatment mouse groups. Note that weights in ICT and ABT + ICT groups include some remaining normal tissue. *n* = 10 UNT, *n* = 6 ABT, *n* = 11 ICT, *n* = 12 ABT + ICT. **d** Higher magnification of H&E-stained

tumor regions shown in (**b**). Dashed line indicates necrotic region in ABT + ICT-treated tumor. Scale bar = 100 μm. **e** Percentage of necrotic area out of tumor lesion area in the different treatment groups. Tissues that did not contain any remaining tumor lesions were not included. *n* = 10 UNT, *n* = 5 ABT, *n* = 9 ICT, *n* = 8 ABT + ICT. **f** Stain of tumor lesions for YFP in same, marking carcinoma cells with lesions. Scale bar = 100 μm. **g** Percentage of lesion area stained for YFP, indicating carcinoma cells, in the different treatment groups. *n* = 8 UNT, *n* = 6 ABT, *n* = 9 ICT, *n* = 8 ABT + ICT. All values indicate mean ± SEM. *P* values were calculated with Brown-Forsythe ANOVA test and Benjamini-Hochberg false discovery rate correction.

observed in the *Ink-ATTAC* mice, yet was somewhat less pronounced. Other central immune cell types did not show significant changes in numbers in the treated mice, and we did not find evidence for an overall inflammatory response (Fig. 7h and Supplementary Fig. 10). Together these findings further indicate that senescent CAFs act to suppress CD8+ T cell activation in PDAC.

**Senescent CAF elimination potentiates immune checkpoint therapy**

The finding that senescent CAF elimination results in increased CD8+ T cell activation rates suggested that senolytic treatment may potentiate treatment with immune checkpoint therapy (ICT). To test this, we implanted the KPC cells in wt mouse pancreata, and treated them with a previously described ICT protocol, which combines anti-CTLA4, anti-PD1, and anti-CD40-agonist antibodies, without additional chemotherapy[41]. Half of the mice also received treatment with the senolytic drug ABT-199, three times a week (Fig. 8a). ABT-199 treatment alone did not affect tumor growth, whereas ICT treatment alone reduced final tumor burden (Fig. 8b, c). However, the combined ICT together with ABT-199 had dramatic effects, with final tumor masses reduced by a two-fold average relative to ICT treatment alone (Fig. 8b, c). Furthermore, the remaining lesions in the mice co-treated with ABT-199 and ICT showed large regions of necrosis, which were not detected to the same extent in the single-treated mice (Fig. 8d, e). Quantification of carcinoma cells within lesion masses by staining for

YFP, labeling the tumor cells, revealed that in the double-treated mice, the proportion of remaining tumor cells within lesions was drastically reduced, indicating that most tumor cells were effectively killed within these lesions (Fig. 8f, g). These results indicate that senolytic treatment targeting mouse tumor stroma can achieve substantially improved tumor response to immunotherapy.

## Discussion

Identification of effective therapeutic approaches to combat PDAC remains a major challenge. Immunotherapies are currently ineffective, and this is attributed to immunosuppression in the stroma of most PDAC tumors[19,20]. There is therefore a great need to uncover the cells and pathways underlying immune suppression within the tumor stroma, and to develop novel methods to tip the balance to favor immune attack of tumor cells. In recent years there have been important advances in the understanding of the complex roles of cellular senescence in cancer. Whereas most recent work has focused on senescence of the tumor cells themselves, occurring either spontaneously or in response to therapy, there is evidence to suggest that different stromal cells can undergo senescence in cancers of various types[8].

In this study, we identify senescent CAFs in the stroma of PDAC tumors and establish a role for these cells in restricting the numbers of cytotoxic CD8+ T cells in the TME. The identification of senescent CAFs is interesting in light of the ongoing research of CAF subtypes in PDAC

as well as other tumor types[19–21,42]. Rather than representing a novel CAF subtype, however, our findings suggest that the senescent state may be acquired by CAFs of different subtypes. This is based on co-expression of senescence markers with markers of different CAF subtypes observed by staining, and on the overlap of the senescence signature in scRNA-seq data with CAF clusters of different subtypes. This suggests that CAFs of various subtypes in the tumor environment may enter a state of senescence, in which they cease to proliferate, activate senescence markers including SA-βGal and the cell cycle inhibitors p16 and p21, and acquire the senescence-associated gene-expression programs. This is in fact consistent with the observation of senescence as a state that can be acquired by distinct cell types, yet encompasses certain universal features[1,2].

Indeed, in investigating the transcriptomes of senescent PDAC CAFs from both mice and humans, we encountered hallmark features of senescent cells. These include reduced expression of proliferation genes, and elevation in stress-response and p53 pathway genes, in ECM components and modifiers, and in cytokines and immune-regulatory genes related to the SASP and to the interferon-response pathway. The specific triggers that induce CAFs to undergo senescence within these developing tumors are unknown. The elevated expression of stress-response gene sets in the senescent CAFs, particularly sets associated with hypoxia, suggest that hypoxic stress may be among these triggers.

To test the functions of senescent CAFs in the context of growing PDAC, we used a newly-developed system to induce senescence of transplanted CAFs, and two systems to eliminate senescent CAFs: the *Ink-ATTAC* model, which relies on elimination of p16+ cells, and the senolytic ABT-199, which inhibits Bcl2. In all three systems results were remarkably consistent, with the most prominent changes observed in the numbers of CD8+ T cells expressing cytotoxic activation markers. Supporting this, treatment of mouse or human T cells in culture with conditioned media from senescent CAFs led to a reduction in CD8+ T cell activation levels.

There is an intense effort to uncover the factors in the TME that repress cytotoxic CD8+ T cell infiltration and activity[19,20,36], and this work highlights senescent CAFs as important players in this interaction. Despite the high inter-regional and inter-tumoral diversity in TME composition in PDAC, we were able to detect a negative correlation between CAF senescence rates and CD8+ T cell numbers and activation in human tumors, supporting these functional findings in the mouse models.

Interestingly, although some of the upregulated cytokines in senescent CAFs, such as Ccl2,3 and Csf1, are known regulators of myeloid-suppressive cells, which are thought to play a major role in restricting T cell activation[19,20,34,36], our experiments did not reveal substantial changes in the numbers of these monocytic populations when senescent CAFs were eliminated or induced. Nevertheless, gene expression and functional features of these cells, not studied here, may also be influenced by the senescent CAFs.

The implanted tumor cells are deficient for p53 and have a silenced p16 gene, and, accordingly, we did not detect senescence in them. Senescent cell elimination by ABT-199 is thus likely restricted to the endogenous stroma, whereas the *Ink-ATTAC* system is also active only in the recipient host stroma. However, although the great majority of stromal senescent cells were CAFs, we cannot rule out the action of these treatments on other senescent stromal cell types. Recent studies have revealed tumor-promoting roles of senescent macrophages in lung adenocarcinoma and of senescent neutrophils in prostate cancer[43–45], and further work will be required to establish whether senescent stromal cells additional to CAFs also play a role in PDAC.

Notably, manipulation of senescent CAF numbers did not substantially influence tumor growth in our experiments, suggesting that the modulation of activated CD8+ T cell numbers by the senescent CAFs is not sufficient to harness substantial tumor cell killing. However, the addition of the senolytic treatment to the ICT drug combination had a clear additional benefit. In particular, it is striking that cancer lesions in the double-treated mice showed a marked decrease in tumor cell content and an increase in necrotic areas, consistent with immune-mediated cancer cell elimination upon treatment. Since ABT-199, venetoclax, is approved for the treatment of several hematopoietic cancers, a combined treatment course for pancreatic cancer may be considered. Overall, this study provides a demonstration that the senolytic elimination of senescent stromal cells could represent a potential approach for enhancing the targeting of tumor cells by the immune system.

## Methods
### Mouse lines
*Ptf1a-CreER* (strain# 019378)[46], C57Bl6, *Kras^{lsl-G12D/+}* (C57Bl6, strain# 008179)[47], and *p53^{flox/+}* (strain# 008462, mixed background)[48] were obtained from Jackson Laboratory and crossed to produce double and triple transgenic lines. Mice were injected subcutaneously at 8 weeks of age with two daily doses of tamoxifen (Sigma) at 400 mg/kg to activate Cre-mediated recombination. Both male and female mice were used in the experiments. Mice were sacrificed 4–10 months after tamoxifen injection for tumor analysis. For bromodeoxyuridine (BrdU) labeling mice were injected intraperitoneally with 100 mg/kg BrdU 2 h prior to sacrifice. The *Ink-ATTAC* transgenic line[39] (C57Bl6) was provided by Sheila Stewart with approval from Unity Biotechnology. Wild-type C57Bl6 mice were purchased from Jackson Laboratory (strain# 00664) and from Envigo. Animals were housed in specific pathogen free conditions, at 20–24 °C, 30–70% humidity and a 12:12 h light-dark cycle. Tumor-bearing mice were monitored 1–3 times a week for tumor growth and health. Tumor size was assessed by palpation, and measured by weight following euthanasia. Tumors did not exceed allowed maximal size of 1.5 g. Mice were terminated when showing signs of distress or illness, including apathy, dehydration, cachexia, cessation of grooming, weight loss of >10%, injury, or abdominal swelling. Mice were euthanized at experimental end points by pentobarbital injection followed by cervical dislocation. The Hebrew University is an AAALAC International accredited institution. All experiments were conducted with approval from the Hebrew University Animal Care and Use Committee.

### Tumor xenografts and treatments
Cell lines derived from PDACs developing in *Pdx1-Cre; Kras^{lsl-G12D/+}; p53^{lsl-R172H/+}* mice were previously described[23,34]. Lines 6422c1 and 6555c3 were labeled with YFP. For xenograft transplantation 3–5 × 10^4 tumor cells were implanted into the pancreata of 6–8-week-old C57Bl6 mice in a 50 μl suspension of 50% Matrigel (Corning) and PBS. Tumors were harvested three weeks after implantation. Both male and female mice were used in the experiments. For co-injection experiments, 3 × 10^4 6555c3 cells were co-transplanted with 2 × 10^6 mouse CAFs transduced with a lentiviral vector expressing the mScarlet fluorescent protein (pLV-mScarlet-I) and carrying vectors for inducible senescence – CreER, and lox-SV40-LT-lox (see below). Mice were treated one and three days after the injection with tamoxifen (Sigma) at 400 mg/kg to activate Cre-mediated recombination, and were sacrificed one week later. Pancreatic regions containing the Matrigel-injected tumor lesions were identified and analyzed. For elimination of p16+ stromal cells in injected *Ink-ATTAC* mice, mice were treated three times a week starting 5 days after tumor cell implantation, with the dimerizing agent AP20187 (WuXi LabNetwork) at 10 mg/kg, dissolved in a 2% Tween20, 10% PEG400, 4% ethanol in water solution, injected intraperitoneally. ABT-199 (MedChemExpress) was administered to mice at 50 mg/kg by oral gavage in 60% phosphal 50 PG (Lipoid), 30% PEG 400 (Sigma) and 10% ethanol. ABT-737 (Wuxi LabNetwork) was administered at 75 mg/kg in 30% propylene glycol (Sigma), 5% Tween 80 (JTBaker) and 3.3% dextrose in water (pH 4–5), injected intraperitoneally.

MCL1i (S63845, MedChemExpress) was dissolved in 25 mM HCl and 20% 2-hydroxy propyl β-cyclo dextrin (Sigma) in saline, and administrated at 40 mg/kg by tail vein injection. ABT-199 treatment to KPC-implanted mice was started 5 days after tumor cell injection. Senolytic treatments of Kras-activated mice were initiated 7-10 months after Kras activation. All senolytic treatments were administrated 3 times a week. Immunotherapy treatments to KPC-implanted mice followed previously-described protocols[34,41], and included three antibodies: anti-PD-1 (clone RMP 1-14, BioXcell), administered 3 times a week for 2 weeks starting 10 days after implantation, at 200 µg per treatment; anti-CTLA4 (clone 9H10, BioXcell), 200 µg, administered 3 times within one week starting 10 days after implantation; anti-CD40 agonist antibody (clone FGK45, BioXcell), 100 µg, injected intraperitoneally once on day 13.

## Patient samples

Patient samples were obtained from the Hadassah-Hebrew University Medical Center, Jerusalem, Israel, Sheba Medical Center Tel-Hashomer, Israel, and from the Midgam, the Israeli Biorepository Network for Research. All experiments were conducted under approval of the institutional Helsinki Committees, with written informed consent, numbers HMO-136-22, HMO-19-0024 (Hadassah) and 5073-18-SMC, 5539-08-SMC (Sheba). All histological sections underwent pathological review to verify lesion type. Patients were aged 35–82 years, with equal representation of males and females. All available patient material was included in the analysis, with no selection. Peripheral blood samples were drawn from healthy adult volunteers with written informed consent at Hadassah Medical Center, under approval of the Hadassah Medical Center Helsinki Committee (number HMO-0024-19). To isolate senescent CAFs, live PDAC specimens from human patient surgeries were minced and enzymatically digested for 45 min at 37 °C with agitation in DMEM 10% FBS, supplemented with 1 mg/ml of Collagenase P (Sigma), 0.5 mg/mL of Liberase DL (Sigma), and 0.2 mg/ml of DNase I (Sigma), and filtered through a 100 µM cell strainer. Cells were then incubated with primary fluorophore-conjugated antibodies (EpCAM, CD45 and CD90), followed by fluorescent SA-βGal stain using $C_{12}$FDG as described below, and Zombie staining for dead cell exclusion. Cells were sorted using a FACS Aria III system (BD Biosciences). Patient sample characteristics are listed in Supplementary Table 1.

## Histological stains and analyses

Tissues were fixed in formalin for 24 h and embedded in paraffin. For immunostaining, 5µm paraffin sections were deparaffinized, rehydrated, underwent antigen retrieval by pressure cooker, and then blocked in CAS-Block (Invitrogen) for 1 h at room temperature. Sections were incubated with primary antibodies overnight at 4 °C, washed and then incubated with secondary antibodies for 30–60 min at room temperature, detected using Peroxidase Substrate Kit (Vector) or fluorescently-labeled secondary antibodies (Jackson), according to standard protocols. Primary and secondary antibodies are referenced in Supplementary Table 3. Bright-field images were collected using an Olympus CX41 and DS-Fi1 camera or with an Aperio AT2 slide scanner (Leica). Fluorescent images were collected using a Nikon C2 Eclipse Ti confocal microscope, a Nikon Ti2A Yokogawa W1 spinning disk confocal microscope, or a Pannoramic Scan II scanner (3DHistech). Quantifications were done by image analysis, either calculating stained areas, or by scoring individual cells, as indicated in figures. Area calculations were done using QuPath or NIS Elements software (Nikon), after defining appropriate positive/negative thresholds. This method was used to calculate stromal and cancer regions, and for defining p16+ stromal areas. An image area of >1 mm² was scored in each tumor. Scoring of individual cell numbers was conducted using QuPath. Total cell numbers in analyzed images were scored using the "cell detection" command, using the DAPI channel in fluorescent images. For each cell-

type marker, positive or double-positive cells were defined according to the following parameters: stain intensity threshold, expression location, cell shape and size, defined in the "select object classifier" command. $1–50 \times 10^3$ individual cells were scored per sample in the analyses in Figs. 1e, j, 4b, d, e, 5d, g, 7d, and S4e; 100–2000 cells in Figs. 1h, 4f, 6d, and S7b. For correlation analyses of human tumors, 57 regions from 11 human PDACs (5 regions in 10 tumors and 7 in 1 tumor) were randomly chosen, verified for histopathological quality and for containing viable stromal and cancer cells. Regions were in the range of 0.2–3.42 mm² area, and contained 832–21235 cells. Consecutive sections were stained for marker combinations that included p16, CD3, CK18, and CD8, GZMB, and VIM. For classification of p16 positive/negative expression in human PDACs, tumor and stromal regions were scored for the presence of p16 cells, with tumors having 0–2 separated regions containing >10 p16+ cells defined as negative, and tumors with 3 or more regions containing >10 p16+ cells defined as positive. Tumor histopathology was analyzed by an expert pathologist (K.A.A.).

## Flow cytometry

For the preparation of single-cell suspensions, excised pancreatic tumors were minced with blades, incubated with Collagenase P (Sigma) and DNAse I (Sigma) for 30 min at 37 °C, and filtered through a 100 µM cell strainer. The cells were incubated with primary fluorophore-conjugated antibodies against surface markers for 30 min on ice in PBS containing 2% FBS. Intracellular staining was performed using the fixation/permeabilization transcription factor buffer set (BD Pharmingen) according to the manufacturer's instructions. Antibodies used in flow analysis are described in Supplementary Table 3. For fluorescent SA-βGal analysis, live cells were incubated with the substrate $C_{12}$FDG (Invitrogen D-2893) at 33 µM for 1.5–2 h at 37 °C. Flow cytometric analysis was performed on an LSR Fortessa flow cytometer (BD Biosciences) and analyzed using FCS Express 7 software.

## Cell isolation and culture

Mouse CAFs were isolated from Kras-driven mouse PDACs. Tumors were digested and single-cell suspensions were prepared by incubation with Collagenase P (Sigma) and DNAse I (Sigma) for 30 min at 37 °C and filtration through a 100 µM cell strainer. CAFs were isolated by FACS using Pdpn+ , Epcam-, CD45-, CD31- labeling and gating. Cell viability staining was performed with Zombie-yellow kit (Biolegend). Staining with antibodies was performed for 30 min at 4 °C. The PDAC primary carcinoma line X252 was established from a primary tumor grown in mice as a patient-derived xenograft, as previously described[49], and was authenticated to the patient's short tandem repeat profile. Mouse CAFs and X252 cells were cultured in RPMI-1640 medium supplemented with 10% FBS, 1% L-glutamine, 1% sodium-pyruvate and 1% pen-strep. Primary human CAF cultures were established from fresh digested PDAC tumors, seeded in culture, with fibroblast outgrowths isolated. KPC cell lines were cultured in DMEM (high glucose without sodium pyruvate) with 10% FBS and 1% L-glutamine.

## Inducible senescence in PDAC CAFs

To generate the inducible senescence system in CAFs, primary isolated human PDAC CAFs were infected with lentiviruses expressing the SV40 large T antigen (LT) and the telomerase catalytic subunit hTERT, flanked by lox sites (loxP-SV40LT and loxP-hTERT). Primary isolated mouse PDAC CAFs were infected directly after sorting with the loxP-SV40LT vector, and both human and mouse CAFs were subsequently infected with the CMV-Cre$^{ERT2}$ vector, carrying a tamoxifen-inducible CreER gene. Standard virus generation procedures were carried out by transfection of the transfer vector, the pHR-ΔR8.2 packaging vector, and the pCMV-VSV-G envelope plasmid, into 293 T cells, with subsequent viral supernatant collection and centrifugal concentration.

Cells were infected with 50 µl of concentrated viral particles in 5-cm dishes, followed by drug selection as needed. For senescence induction, cells were treated with 1 µM 4-hydroxytamoxifen (4-OHT; Sigma) for 10 days to activate the CreER construct. For SA-βgal staining, cells were fixed with 0.2% glutaraldehyde, followed by incubation at 37 °C with 200 mM citric acid/sodium-phosphate buffer, 5 mM K4[Fe(CN)6]3H2O, 5 mM K3[Fe(CN)6], 2 mM MgCl$_2$, 150 mM NaCl and 1 mg/ml X-gal (Inalco) in PBS pH = 5.8. For BrdU staining, cells were seeded in chamber slides (ibidi) and fixed with 4% paraformaldehyde, permeabilized with 0.5% Triton X100 in PBS, blocked with CAS-Block and 2.5% BSA for 10 min and stained with primary antibodies for 1 h, followed by fluorophore-conjugated secondary antibodies and DAPI nuclear stain. Fluorescent images were collected using a Nikon C2 Ti Eclipse confocal microscope and processed with NIS-Elements software. IFN-γ treatments were conducted by adding mouse recombinant IFN-γ (PeproTech) to mouse CAFs at a concentration of 1000 u/ml for 48 h. For conditioned media collection from CAFs, cells were washed in PBS, and then cultured for 48 h in serum-free media. Conditioned media were then collected and filtered through a 0.45 µm filter. For the proliferation assay of X252 human PDAC cells, the cells were seeded in 96-well plates (Nunc) at 2500 cells/well, and were treated with conditioned media from non-senescent or senescent human CAFs at a 1:1 ratio with fresh media. Phase-contrast images were taken every 24 h over 4 days using the Incucyte S3 Live-Cell Analysis instrument (Sartorius). Confluence metric data were obtained from the Incucyte analysis software (version 2019B Rev2). For tumorsphere growth, Master 3D Petri Dish 35-well arrays (Microtissues Inc., RI, USA), were used to create agarose hydrogel microwells into which cells were seeded to form spheroids. Mouse CAFs were labeled with mScarlet and seeded together with YFP-labeled 6555c3 KPC cells at a 4:1 ratio, as 4000 cells per spheroid in RPMI supplemented with 10% FBS, 1% pen-strep, 1% sodium-pyruvate and 1% L-glutamine, and grown for 6 days. Fluorescent images were collected using a Nikon TL microscope. Quantification of the YFP+ area within the spheres was done using the NIS Elements software. All cell lines used in this study were routinely tested for mycoplasma contamination.

## T cell assays

Human peripheral blood mononuclear cells (PBMCs) were isolated from healthy donors with Lymphoprep purification (Serum Bernburg AG), according to manufacturer's instructions. PBMCs were cultured in RPMI supplemented with 10% FBS, 1% pen-strep, 1% L-glutamine, non-essential amino acids, 25 mM HEPES and 50 µM β-mercaptoethanol, and were activated for a week with human IL2 (Peprotech) at 6000 u/ml, and anti-CD3 (BioLegend; OKT3; 25 ng/ml). For testing of T-cell activation state, cells were re-stimulated in plates pre-coated with anti-CD3 (Biolegend; OKT3; 1µg/ml) for 6 h together with exposure to conditioned media. After two hours BFA (TONBO Biosciences) was added to each well. T cells were surface-labeled with antibodies against CD8, CD4 and CD25, and intracellular staining for IFN-γ and Granzyme B was performed using fixation/permeabilization transcription factor buffer set (BD Pharmingen), and samples were analyzed by flow cytometry. To assess secreted IFN-γ levels, cells were re-stimulated with pre-coated anti-CD3 (Biolegend; OKT3; 125 ng/mL) together with exposure to conditioned media in a 1:1 ratio with fresh media. After 18 h, media were collected and IFN-γ levels were measured by ELISA (Biolegend). Mouse splenocytes were extracted from pmel-1 mice and activated with gp100 peptide at 1 µg/ml (A2S; Genscript). Cells were cultured in RPMI supplemented with 10% FBS, 1% pen-strep, 1% L-glutamine, non-essential amino acids, 25 mM HEPES and 50 µM β-mercapto-ethanol, with human IL2 at 400 u/ml. After one week, cells were re-stimulated in 96 well plates pre-coated with anti-CD3 (Biolegend; 1 µg/ml) and anti-CD28 (Biolegend; 0.5 µg/ml) for 6 h, together with exposure to conditioned media (in 1:1 ratio with fresh media) obtained from mouse CAFs

either untreated or stimulated with IFN-γ for 48 h prior to washing and conditioned media collection.

## RNA extraction, qRT-PCR, and transcriptome analysis

RNA extraction from FACS-sorted SA-βGal-stained cells was performed using the RNA isolation kit Direct-zol RNA Microprep (Zymo Research), using >5000 cells of each sample. RNA was extracted from cultured CAFs using the RNA isolation kit Direct-zol RNA Miniprep (Zymo Research). cDNA was synthesized using the iScript cDNA synthesis kit (Bio-Rad) and qRT-PCR was performed using iTaq Universal SYBR Green Supermix (Bio-Rad) in triplicate reactions, with expression values normalized to GAPDH and β-actin expression levels for mouse and human cells. Mouse p16 mRNA levels were measured by TaqMan. Primer lists are provided in Supplementary Table 3. For transcriptome profiling, mRNA from senescent and non-senescent cells was extracted and analyzed using CEL-seq2. 3' cDNA was synthesized and barcoded, followed by RNA synthesis, amplification by in vitro transcription, and library generation for paired-end sequencing. Reads were demultiplexed, quality-filtered, and trimmed for adapters and poly-A tail using Cutadapt, aligned with the mouse genome (GRCm38) or human genome (HSA GRCh38) using Tophat2, and counted using Salmon v0.13.1. Differential gene expression was conducted using DESeq2 with batch correction, and enrichment for up- and down-regulated gene sets was conducted by Metascape[50] and by GSEA[51], testing gene sets from the hallmark collection of the molecular signatures database (MSigDB).

## scRNA-seq analysis

Single-cell mRNA-seq data of human pancreatic tumors were obtained from ref. 31. The dataset was pre-filtered and cell identities were provided in metadata format. Tumor sample data were extracted using the Subset function of Seurat v4 and were normalized and integrated using the FindIntegrationAnchors function, followed by the IntegrateData function dimensions, after selecting 2000 variable genes for each sample (using the FindVariableFeatures function). Fibroblast cell data was taken for further analyses and was filtered out of low-quality samples, ending with 3492 cells from 21 tumor samples. To identify CAF subtypes, signature scores were calculated by the AddModuleScore function. The signature score (SC) is the average scaled expression ($E_s$) of a gene set (G) subtracted by aggregated expression of the control gene set (C), to control for confounding cell complexity: SC=average[$E_s$(G)] – average[$E_s$(C)]. A control gene set was matched to each gene set to represent comparable expression distribution. For each gene of a given gene set 100 control genes were randomly selected from the same expression bin. The results were visualized by FeaturePlot function. Signatures of iCAFs, myCAFs and apCAFs were obtained from ref. 23. and scores were calculated using the addModuleScore function. For the calculation of senescence percentage in each CAF subtypes, individual cells were classified into one of the subtypes if they displayed a ModuleScore>0 for only one subtype signature. Cells positive for multiple subtype signatures or none were not included in this calculation. Individual CAF marker genes were visualized in FeaturePlot as single gene expression values. To define senescent CAFs, we modified the senescent index tool (SIT) algorithm developed by ref. 32. calculating the following: *Senescence Score = Senescence signature$_{norm(0-1)}$ – Cell cycle score$_{norm(0-1)}$* with (*norm(0-1) – 0-1 scaling of scores using scale*). The Senescence signature was defined as the scaled (0-1) signature score of several senescent markers: *CDKN2A, CDKN2B, CDKN1A, CDKN1B* and *SERPINE1*. The Cell Cycle signature was defined as the scaled (0-1) signature score of 53 cell cycle genes common to 3 different cell cycle gene sets (KEGG, REACTOME and WP CELL CYCLE). Cells with a Senescence Score>0 were defined as senescent. We used MAST to identify differentially-expressed genes between senescent CAFs (Senescence Score>0) and non-senescent CAFs (Senescence Score ≤0), using Seurat FindMarkers. The top

positively differentially expressed genes (avg_Log2FC > 0 and Bonferroni $P$adj values ≤ 0.2) were tested for gene set enrichment using Metascape. Genes included in the signatures used are provided in Supplementary Table 2.

## Mass spectrometry

Conditioned media (serum free) was collected from non-senescent and senescent human CAFs after 24 h of incubation, and centrifuged to remove cell remnants and debris. Four independent replicate samples were collected. The medium samples were precipitated, and washed 2 times with cold 80% acetone. The protein pellets were dissolved in 8.5 M Urea, 400 mM ammonium bicarbonate and 10 mM DTT. Protein amount was estimated using Bradford readings. The samples were reduced (60 °C for 30 min), modified with 35.2 mM iodoacetamide in 100 mM ammonium bicarbonate (room temperature for 30 min in the dark) and digested in 1.5 M Urea, 66 mM ammonium bicarbonate with modified trypsin (Promega), overnight at 37 °C in a 1:50 (M/M) enzyme-to-substrate ratio. An additional second trypsinization was done for 4 h. The resulting tryptic peptides were analyzed by LC-MS/MS using Exploris 480 mass spectrometer (Thermo) fitted with a capillary HPLC (Evosep one). The peptides were loaded onto a 15 cm ID 150 1.9-micron (Batch no. E1121-3-24) column of Evosep. The peptides were eluted with the built-in Xcalibur 15 SPD (88 min) method. Mass spectrometry was performed in a positive mode using repetitively full MS scan (m/z 350–1200, resolution 120,000 for MS1 and 15,000 for MS2) followed by high collision dissociation (HCD, at 27 normalized collision energy) of the 20 most dominant ions (>1 charges) selected from the full MS scan. A dynamic exclusion list was enabled with exclusion duration of 20 s. The mass spectrometry data was analyzed using the MaxQuant software 2.1.1.0. (//www.maxquant.org) using the Andromeda search engine, searching against the proteome from the human Uniprot database with mass tolerance of 4.5 ppm for the precursor masses and 4.5 ppm for the fragment ions. Peptide- and protein-level false discovery rates (FDRs) were filtered to 1% using the target-decoy strategy. Protein table was filtered to eliminate the identifications from the reverse database, and common contaminants. The data was quantified by label free analysis using the same software, based on extracted ion currents (XICs) of peptides enabling quantitation from each LC/MS run for each peptide identified in any of experiments. Statistical analysis of the identification and quantization results was done using Perseus 1.6.7.0 software.

## Statistical analyses

Unpaired Student's $t$ tests or Chi-squared tests were used to compare experimental groups as indicated. All $t$ tests were two-sided. $P$ values of correlations were calculated using Spearman's or Pearson correlation as indicated. For multigroup comparisons, $P$ values were calculated with Brown-Forsythe ANOVA test and Benjamini-Hochberg false discovery rate correction.

For analysis of transcriptomic data, DESeq2 was used to calculate gene expression fold changes. Gene set enrichment values were calculated using Metascape and GSEA, and with MAST for scRNA-seq data. $P$adj values indicate multiple hypothesis correction using the Bonferroni method (Metascape), or the Benjamini-Hochberg false discovery rate method (DESeq2 and GSEA).

## Reporting summary

Further information on research design is available in the Nature Portfolio Reporting Summary linked to this article.

## Data availability

The mRNA-seq expression profiles generated in this study have been deposited in the Gene Expression Omnibus (GEO) database under accession number GSE235246. The mass spectrometry data have been deposited to the ProteomeXchange Consortium via the PRIDE partner repository with the dataset identifier PXD053144. The scRNA-Seq data generated by others and analyzed in this study is available at the Genome Sequence Archive (GSA) under accession number CRA001160. Source data are provided with this paper.

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

## Acknowledgements

We thank Philippe Ravassard for the lentiviral constructs of hTERT, LT and CreER, David Tuveson for KPC cells, Nabeel Bardeesy for tumor sections, Shai Carmi for assistance in statistical analysis, Jonathan Monin for bioinformatics analysis, Norma Kidess-Bassir for histological preparation, Hiba Mohalwes and Raghad Herbawi for mouse genotyping and technical analyses, Hasan Sourikh for slide scanning, Yael Feinstein and Zakhariya Manevitch for assistance with microscopy, Dan Lehman for assistance with FACS, and Tamar Ziv and the Smoler Proteomics Center at the Technion for assistance with mass spectrometry. This study was supported by grants from the Israel Science Foundation - Broad Institute program (2621/18, I.B.-P.), Israel Precision Medicine Partnership (3755/21, I.B.-P.), Israel Science Foundation (820/22, I.B.-P.), Israel Science Foundation Mid-Career Program (1923/22, I.B.-P.), the Israel Ministry of Science and Technology DKFZ-MOST program (4062, I.B.-P.), the Chief Scientist of the Israel Ministry of Health (3-15017, I.B.-P.), the Alex U. Soyka Program (I.B.-P., B.A., R.K., L.H.), the Israel Cancer Research Fund International Collaboration Program (I.B.-P.), the Israel Cancer Research Fund and Cancer Research Institute (M.B.), NIH grants R01 AG059244, CA217208 (S.A.S.), U.S. Army Medical Research Acquisition Activity BC181712 (S.A.S.), the Siteman Cancer Center Investment Program (S.A.S.), NIH grant K99-R00 CA252153 (J.R.P.), NIH grants R01-CA-252225. R01-CA-276512 (B.Z.S.), the Gassner Fund for Medical Research (B.A.), the Teva Israeli Bioinnovators Fellowship (B.A.), the Abisch-Frenkel Foundation Fellowship (L.H.) and the Kaye Einstein Scholarship (L.H.). I.B.-P holds the Woll Sisters and Brothers Chair in Cardiovascular Diseases. Diagrams were generated using BioRender.com under Academic License terms.

## Author contributions

B.A., R.K., and L.H. designed the study, conducted experiments and prepared the manuscript. R.C., and T.M.-H. conducted experiments. Y.E. and M.B. assisted in design and execution of T cell experiments. E.E., J.R.P., and B.Z.S. contributed KPC cell lines and tumors and assisted in execution and analysis of tumor xenograft and ICT experimentation. D.A. and T.G. contributed primary human PDAC cells and CAFs and assisted in CAF experimentation. S.A.S. contributed mice and experimental advice. E.S. and O.B. assisted in design and execution of tumorsphere experiments. T.B.-M. conducted section scanning. E.F., L.H.K., K.A.A., and G.Z. assisted in obtaining human PDAC sections and surgical samples, and K.A.A. conducted pathological analyses. M.H. assisted in scRNA-seq analysis. I.B.-P. supervised the study and wrote the manuscript.

## Competing interests

B.Z.S. receives sponsored research support from Boehringer-Ingelheim and Revolution Medicines and holds equity in iTeos Therapeutics. T.G. Receives research support from Astra Zeneca, Abbvie honoraria, consultation fees from Abbvie and MSD Merck, royalties and consultant fees from Curesponse, and speaker fees from Abbvie and ClearNote Health. The remaining authors declare no competing interests.

## Additional information

[1]Department of Developmental Biology and Cancer Research, Institute for Medical Research Israel-Canada, Faculty of Medicine, The Hebrew University of Jerusalem, Jerusalem, Israel. [2]The Lautenberg Center for Immunology and Cancer Research, Institute for Medical Research Israel-Canada, Faculty of Medicine, The Hebrew University of Jerusalem, Jerusalem, Israel. [3]Department of Biochemistry, Institute for Medical Research Israel-Canada, Faculty of Medicine, The Hebrew University of Jerusalem, Jerusalem, Israel. [4]Department of Surgery, Hadassah Medical Center, and Faculty of Medicine, Hebrew University of Jerusalem, Jerusalem, Israel. [5]Division of Hematology-Oncology, Department of Medicine, University of Massachusetts Chan Medical School, Worcester, MA, USA. [6]Department of Molecular Cell and Cancer Biology, University of Massachusetts Chan Medical School, Worcester, MA, USA. [7]Pancreatic Cancer Translational Research Laboratory, Oncology Institute, Sheba Medical Center, Tel Hashomer, Israel, and Sackler Faculty of Medicine, Tel Aviv University, Tel Aviv, Israel. [8]Institute for Drug Research, School of Pharmacy, Faculty of Medicine, Hebrew University of Jerusalem, Jerusalem, Israel. [9]Department of Gastroenterology, Hadassah Medical Center, and Faculty of Medicine, Hebrew University of Jerusalem, Jerusalem, Israel. [10]The Rachel and Selim Benin School of Computer Science and Engineering, The Hebrew University of Jerusalem, Jerusalem, Israel. [11]Department of Cell Biology and Physiology, Washington University School of Medicine, St. Louis, MO, USA. [12]Department of Pathology, Hadassah Medical Center, and Faculty of Medicine, Hebrew University of Jerusalem, Jerusalem, Israel. [13]Department of Medicine and Abramson Family Cancer Research Institute, Perelman School of Medicine, University of Pennsylvania, Philadelphia, PA 19104, USA. [14]These authors contributed equally: Benjamin Assouline, Rachel Kahn, Lutfi Hodali. ✉e-mail: ittaibp@mail.huji.ac.il

