## [Transparent Peer Review file · Nature Communications]

Senescent cancer-associated fibroblasts in pancreatic adenocarcinoma restrict CD8⁺ T cell activation and limit responsiveness to immunotherapy in mice

Corresponding Author: Professor Ittai Ben-Porath

Version 0:

Reviewer comments:

Reviewer #1

(Remarks to the Author)

In this manuscript by Assouline and colleagues, the authors investigate the significance of senescence among cancer-associated fibroblasts in the pancreatic tumor microenvironment. This study addresses two features of solid tumors with somewhat controversial roles in tumor progression—senescence and CAFs—and does so using rigorous approaches and careful analyses, thus poised to move the field forward. The work is also novel, as senescence has been investigated functionally within the CAF compartment of other tumor types but has been subject to little study in the context of pancreatic cancer previously. By analyzing CAFs positive versus negative for senescence markers, the authors have identified a CAF state as opposed to highly plastic transcriptional program, which has enabled them to manipulate these cells in vivo and assess their functions. Functional studies of CAFs within their relevant tissue environments are largely lacking from the literature, and points to another strength of this manuscript. Weaknesses are modest, but some further evidence in support of the proposed functional interaction between senescent CAFs and exclusion or inactivation of cytotoxic T cells would help to strengthen the manuscript.

Specific comments:

1. The results depicted in Figure 3c-l reflect very short outgrowth of tumor cells and CAFs in vivo when tumors are very small and likely not reflective of the immune-suppressive and fibrotic microenvironment characteristic of large, invasive tumors. The consequences of CAF senescence in this setting only 1-3 days after implantation may therefore be quite different than in a large established tumor. As the implanted CAFs are immortalized, it seems reasonable to expect that they will be retained in the tumor microenvironment over the course of disease progression, such that phenotypes may well be evident using this senescence gain-of-function approach despite the presence of CAFs from pancreas-resident sources. It would thus be important to repeat these experiments injecting TAM to induce stromal senescence when tumors are established (several weeks after implantation) and key endpoints such as immune cell abundance analyzed, to complement the results of very early senescence induction presently shown.
2. While the senescence induction system works as expected in murine CAFs in culture as demonstrated in Figure 2, it is important to demonstrate that this system works as expected and that TAM induces CAF senescence in vivo to accompany the results in Figure 3, to ensure appropriate interpretation, as cues from neighboring cell types may potentially perturb the induction of senescence in vivo.
3. While the impact of senescent CAF depletion in Figure 4e, f, k, and l on CD8 T cells is interesting, it seems possible that this is at least in part a reflection of an inflammatory response to cell death among a prominent proportion of stromal cells, as opposed to an implication that senescent CAFs restrain CD8 T cell abundance per se. As dead, previously senescent CAFs are likely cleared from tumor tissue within a few days, it would be informative for the authors to report CD8 T cell abundance and/or activation marker profiles at a later time point after the final treatment with AP, assuming senescent CAFs are still reduced in abundance.

Reviewer #2

(Remarks to the Author)

Senescent cancer-associated fibroblasts in pancreatic adenocarcinoma suppress CD8⁺ T cell activation and inhibit response to immunotherapy
Assouline et al.

In this study, the authors examine the effect of the presence of senescent CAFs within the PDAC microenvironment. The

manuscript illustrates that senescent CAFs have an important contribution in lowering CD8+ T cells activation, and their removal (genetically or through treatment) leads to an increased presence of activated CD8+ T cells. Moreover, the combination of a senolytic drug together with different ICBs was proven to improve the treatment response compared to the ICBs alone, in mice.

The authors use different methods in vitro and in vivo, such as mouse models, human samples, RNA seq and computational tools in order to prove the presence and the effect of senescent CAFs in the PDAC microenvironment.

Overall the manuscript is well written, the link between senescent CAFs and t cell activation is interesting and the potential use of senolytic drugs is promising. Nevertheless, the novelty is somewhat limited by a large body of previous evidence linking CAFs and particularly iCAF with senescence and SASP. The link to T cell activation is interesting and adds new insights, but is not strong enough, at this point. The authors should solidify the significance of the manuscript by supporting their claims with more human data showing the link between T cell inhibition and senescent CAFs, more analysis of the CAF subsets, and more rigorous image analysis showing the CAF-specific effects.

Major

- Senescent cells, and specifically senescent fibroblasts, are well known to be present in stromal PDAC, and therefore the novelty of Figure 1 is limited. The main novelty proposed is in Figure 1e, where the authors show senescence in different subtypes of CAFs, and using PDGFRa as an iCAF marker. However PDGFRa is considered as a general CAF marker rather than an iCAF one, and the subset characterization of CAF subsets should be solidified using more markers. I would also suggest reconsidering moving parts of this figure to the supplementary material. Moreover, most of the Figure completely lacks quantification of the imaging work which must be included. The quantification shown in the supplementary Figure should be in a format that provides statistical significance testing.
- Following up on this point, senescent signatures (i.e. SASP) are well known to correspond with iCAF signatures, limiting the novelty of Figure 2.
- Figure 3a-b – The correlation is not very strong, and given that a majority, but certainly not all senescent cells are CAFs (only ~60% of senescent cells in the TME are CAFs), the authors should support the correlation between senescent CAFs and CD3 staining by costaining for CAF markers (i.e correlate PDPN+P16+ cells with presence of CAD3+ cells) in order to prove that the effect seen due to senescence is really coming from CAFs and not from other cells. This is particularly important since in Fig 3i the authors show that the total number of CD3+ cells does not change following senescence activation in the injected CAF system. Following up on this point, this is minor, however what is the other population expressing senescent markers in this model (the remaining 30% in Supp Figure B)? This could potentially be another stromal population (pericytes? Endothelial cells? The authors do not mention this) that could induce similar effects.
- Following up on the previous point, in Figure 4 once again the model removes all senescent cells in the mouse, not just the senescent fibroblasts, so the results may be due to other cells. Moreover, the effects are modest and no difference in tumor progression is shown.
- Figure 5 - I appreciate the proof of concept of the combination of the senolytic drug with ICB therapy, however I wonder whether this harsh therapeutic combination is durable and well tolerated.
- The authors should perform mass spectrometry on the CM of the senescent CAFs in order to unravel the mechanisms behind the effect they have on T cells activation.
- The authors should stain human samples for senescent CAF markers together with markers of CD8+ T cells to validate that the observed phenotype has human relevance.

Minor

- Figure 1c - what are the numbers on top of the bars of the graph? It is not mentioned in the figure legend and it doesn't seem like the average value.
- Figure 1d - row 19 page 6 "Stromal p16 expression was often observed in regions surrounding the epithelial lesions, and the majority of p16+ cells co-localized with expression of the CAF marker Pdpn2" - the sentence needs to be supported by quantification: number of samples and measure of proximity or colocalization.
- Figure 1e - quantification is missing.
- Figure 1f - the colors of the writing don't match the colors of the image and it can be confusing for some readers.
- Figure 1i - the number of replicates is not specified.
- Row 20 page 7: "Most sections containing human premalignant lesions showed the presence of p16+ cells in the epithelial as well as stromal compartments (8 of 17)" - The authors should rephrase this sentence since 8 of 17 is not "most".
- Supplementary Figure 1d – what is the threshold to define the area positive or negative for p16+? It should be elaborated in the methods or in the figure legend.
- The authors may wish to reconsider the organization of figure 2: instead of having in the same figure PanIn mouse and PDAC human perhaps insert in the main figure the RNAseq of PDAC mouse (switch between supplementary and main).
- Supplementary figure 4c-d: it is claimed that the p16+ stromal regions trend towards having lower CD3+ content, but the representative images shown are misleading since they contain a large amount of cancer areas.
- Figure 3a: scale bars are not defined. The four pictures seem in different magnifications.
- Page 13 row 9 "senescent cell elimination is restricted to the stroma" - it is restricted in all cells of the mouse, not just in the stroma. I suggest rephrasing.
- Figure 5b - increase quality of the image.

Reviewer #3

(Remarks to the Author)

This is an excellent paper characterizing the role of senescent CAFs in the immunogenicity of pancreatic cancers. I basically just want to congratulate the authors for this complete, original, elegant and relevant investigation.

I don't really have any criticism, but a couple of curiosities that authors may want to consider:

1. Fig. 3b. This figure shows a negative correlation between p16+ CAFs and intratumoral CD8. I wonder if this negative correlation is specific for p16+ CAFs or if it also true for p16- CAFs or for CAFs in general.
2. Figs. 3j,k,l: These figures demonstrate the inhibitory effect of the SASP on CD8 cells in vitro. I wonder what would be the effect if CAFs and CD8 cells are able to interact? This would recapitulate closer the situation in vivo. Senescent cells have been reported to express immunosuppressive and immunostimulatory cell surface ligands.
3. I guess a prime candidate factor from the SASP to mediate the immunosuppression of CD8 responses is TGF β (which they also find upregulated in their senescent CAFs). A comment on this would be appropriate. Have authors tried galunisertib in their in vitro assays? (this does not seem too complicated and it would be a nice addition to their already very complete paper).
4. Regarding Fig. S7: I found interesting the high levels of BclXL in the tumoral compartment and the depletion of p16+ cancer cells by navitoclax. Perhaps this is worth mentioning.

Author Rebuttal letter:

RESPONSE TO REVIEWERS' COMMENTS

Reviewer #1 (expert in pancreatic cancer):

In this manuscript by Assouline and colleagues, the authors investigate the significance of senescence among cancer-associated fibroblasts in the pancreatic tumor microenvironment. This study addresses two features of solid tumors with somewhat controversial roles in tumor progression—senescence and CAFs—and does so using rigorous approaches and careful analyses, thus poised to move the field forward. The work is also novel, as senescence has been investigated functionally within the CAF compartment of other tumor types but has been subject to little study in the context of pancreatic cancer previously. By analyzing CAFs positive versus negative for senescence markers, the authors have identified a CAF state as opposed to highly plastic transcriptional program, which has enabled them to manipulate these cells in vivo and assess their functions. Functional studies of CAFs within their relevant tissue environments are largely lacking from the literature, and points to another strength of this manuscript. Weaknesses are modest, but some further evidence in support of the proposed functional interaction between senescent CAFs and exclusion or inactivation of cytotoxic T cells would help to strengthen the manuscript.

We thank the reviewer for this supportive assessment.

Specific comments:

1. The results depicted in Figure 3c-l reflect very short outgrowth of tumor cells and CAFs in vivo when tumors are very small and likely not reflective of the immune-suppressive and fibrotic microenvironment characteristic of large, invasive tumors. The consequences of CAF senescence in this setting only 1-3 days after implantation may therefore be quite different than in a large established tumor. As the implanted CAFs are immortalized, it seems reasonable to expect that they will be retained in the tumor microenvironment over the course of disease progression, such that phenotypes may well be evident using this senescence gain-of-function approach despite the presence of CAFs from pancreas-resident sources. It would thus be important to repeat these experiments injecting TAM to induce stromal senescence when tumors are established (several weeks after implantation) and key endpoints such as immune cell abundance analyzed, to complement the results of very early senescence induction presently shown.

In this KPC transplantation model tumor growth is quite rapid → we transplant 30K cells, and mice must be sacrificed 3 weeks subsequently due to large tumor growth and morbidity. The experiments shown in Figure 3c-l (now Figure 4a-g) were conducted one week after transplantation, with TAM treatment administered 1-3 days after transplantation to induce CAF senescence over this week of growth. Thus, while these are small tumors, they do not substantially differ from the tumors found 1-2 weeks subsequently.

We attempted to conduct the experiment, as suggested, by inducing senescence in the co-transplanted CAFs at a later stage that would allow analysis of larger tumors. However, despite testing various experimental configurations, we unfortunately were not able to overcome the main problem (noted by the reviewer): as the tumors rapidly grow to encompass many millions of cells, the originally injected CAFs are diluted out, such that in the formed tumors they are detected in small numbers or not at all. This is because the CAFs, even prior to senescence induction, do not proliferate *in vivo* at a rate that matches that of the tumor cells. Consequently, the effects of the co-injected CAFs in this setting are not detectable at this later stage. We now added text to indicate this experimental limitation of the co-injection system (p.13 rows 10-12).

Nevertheless, we believe that the setting of the small lesions allows a controlled co-transplanted environment which allows observing the effects of CAF senescence induction. Importantly, the experiments involving senescent CAF elimination shown in Figures 5-7 were all conducted in tumors grown for 3 weeks to reach larger sizes, and therefore the findings in the two experimental systems support each other.

2. While the senescence induction system works as expected in murine CAFs in culture as demonstrated in Figure 2, it is important to demonstrate that this system works as expected and that TAM induces CAF senescence *in vivo* to accompany the results in Figure 3, to ensure appropriate interpretation, as cues from neighboring cell types may potentially perturb the induction of senescence *in vivo*.

We now add stains of the CAFs after co-injection with the tumor cells, which show that following tamoxifen treatment there is loss of T-antigen expression, indicating successful recombination, accompanied by elevation of p21 expression and reduced proliferation of these CAFs. This indicates that indeed the system works *in vivo*, inducing senescence in some if not all injected CAFs. This is now described in the text (p.12-13) and in new Figure S6a,b.

These stains were in fact quite challenging as they required co-staining for the mScarlet marker of the injected CAFs, together with additional markers for which antibodies are quite poor. Please also note that p16 cannot be used as a senescence marker in this system, as it acts upstream to Rb and therefore is elevated in the engineered CAFs prior to large T-antigen removal. We believe this important point is now convincingly shown in the revised manuscript.

3. While the impact of senescent CAF depletion in Figure 4e, f, k, and l on CD8 T cells is interesting, it seems possible that this is at least in part a reflection of an inflammatory response to cell death among a prominent proportion of stromal cells, as opposed to an implication that senescent CAFs restrain CD8 T cell abundance *per se*. As dead, previously senescent CAFs are likely cleared from tumor tissue within a few days, it would be informative for the authors to report CD8 T cell abundance and/or activation marker profiles at a later time point after the final treatment with AP, assuming senescent CAFs are still reduced in abundance.

To address this question we conducted an experiment in which we treated tumor bearing Ink-ATTAC mice with AP for one week to eliminate the senescent CAFs, and then stopped treatment for another week, to test whether the prior elimination would suffice to change CD8+ T cell activation and abundance. However, we found that this resulted in reappearance of CAF senescence by the time the mice were sacrificed, and reduction in the observed effect on the T cells. This is not surprising as it appears that these tumors generate constant signals to induce CAF senescence, but precludes answering the question raised by the reviewer in this manner.

We therefore whether this question can be addressed in another way. We wish to point out that both systems used for the elimination of the senescent CAFs by definition rely on induction of apoptosis, which is highly rapid, and non-inflammatory. The Ink-ATTAC transgenic system induces apoptosis through activation of Caspase 8, which is known to actively block inflammatory death pathways. The senolytic treatment blocks Bcl2, similarly inducing apoptosis. Cell elimination is therefore

rapid, and we believe it is unlikely that cell fragments induce residual inflammation. Senolytic treatment is in fact most often associated with decreased inflammation, due to removal of SASP-secreting senescent cells.

To study whether there is evidence of ongoing inflammation following senescent cell elimination from tumors we stained sections for total immune cells (Cd45), T cells, and monocytes, and found that there was not a significant increase in these cell types, nor did we observe clusters of immune infiltrates indicating an inflammatory response. We also stained the treated tumors for nuclear p65 RelA, the NFkB subunit, and for phospho-Stat1, both markers of inflammation. These stains similarly did not reveal a change in expression pattern in the treated versus untreated tumors, in both models. This data is now shown in new Figures S8 and S10 and noted in the text (p.15 rows 7-9, and p.16 rows 12-13). Altogether these data, while not formally excluding the query raised by the reviewer, argue against this possibility. Reviewer #2 (expert in cancer-associated fibroblasts):

Senescent cancer-associated fibroblasts in pancreatic adenocarcinoma suppress CD8+ T cell activation and inhibit response to immunotherapy
Assouline et al.

In this study, the authors examine the effect of the presence of senescent CAFs within the PDAC microenvironment. The manuscript illustrates that senescent CAFs have an important contribution in lowering CD8+ T cells activation, and their removal (genetically or through treatment) leads to an increased presence of activated CD8+ T cells. Moreover, the combination of a senolytic drug together with different ICBs was proven to improve the treatment response compared to the ICBs alone, in mice. The authors use different methods in vitro and in vivo, such as mouse models, human samples, RNA seq and computational tools in order to prove the presence and the effect of senescent CAFs in the PDAC microenvironment. Overall the manuscript is well written, the link between senescent CAFs and T cell activation is interesting and the potential use of senolytic drugs is promising. Nevertheless, the novelty is somewhat limited by a large body of previous evidence linking CAFs and particularly iCAF with senescence and SASP. The link to T cell activation is interesting and adds new insights, but is not strong enough, at this point. The authors should solidify the significance of the manuscript by supporting their claims with more human data showing the link between T cell inhibition and senescent CAFs, more analysis of the CAF subsets, and more rigorous image analysis showing the CAF-specific effects.

We thank the reviewer for these comments, which are addressed below. The revised manuscript contains new data providing further details about senescence and CAF subtypes, demonstration of CAF-specific elimination based on image analysis, and analysis of human samples.

Major

- Senescent cells, and specifically senescent fibroblasts, are well known to be present in stromal PDAC, and therefore the novelty of Figure 1 is limited. The main novelty proposed is in Figure 1e, where the authors show senescence in different subtypes of CAFs, and using PDGFRa as an iCAF marker. However PDGFRa is considered as a general CAF marker rather than an iCAF one, and the subset characterization of CAF subsets should be solidified using more markers.

In the revised version we provide a better characterization of the association of CAF markers with p16. Our analysis indicates that Pdpn provides the most general staining of CAFs, by either FACS or IHC. FACS analysis of transgenic Kras-driven lesions showed that Pdgfra+ cells represent approximately 45% of Pdpn+ cells. This is now shown in new Figure S1d,e. We also show by section staining and image analysis that out of stromal p16+ cells, the largest proportion of cells are indeed Pdpn+ (new Figure 1e), i.e. CAFs. We co-stained lesions for p16 together with Pdgfra and aSma, and found that p16+ cells are detected among CAFs expressing either marker (new Figure 1h). Interestingly, we found p16+ cells also among double positive Pdgfra+Sma+ CAFs, indicating that these markers do not mark fully distinct populations in these Kras-driven lesions. While several previous publications have associated Pdgfra expression preferentially with iCAFs, we modified the text to reflect the reviewer's comment (p.6-7) We also provide additional information in the section analyzing the human scRNA-seq data, showing multiple markers of CAF subtypes (Figure S4c). Overall these analyses support our contention that CAFs of different subtypes can undergo senescence.

I would also suggest reconsidering moving parts of this figure to the supplementary material. Moreover, most of the Figure completely lacks quantification of the imaging work which must be included.

The quantification shown in the supplementary Figure should be in a format that provides statistical significance testing.

We now added new quantification graphs to cover all of the panels in Figure 1:

1e â quantification of the % of p16+ cells that are CAFs, i.e. Pdpn+

1f â quantification of distance of p16- and p16+ CAFs to nearest cancer cell.

1h â quantification of p16+ CAF subsets, co-stained for Pdgfra and Sma

1j â quantification of BrdU+ cells among p16- and p16+ Pdpn+ CAFs

1n â quantification and statistics for the human stained samples (previously in supplementary data, referred to in the last comment by the reviewer).

- Following up on this point, senescent signatures (i.e. SASP) are well known to correspond with iCAF signatures, limiting the novelty of Figure 2.

We agree with the reviewer that there is indeed overlap between the inflammatory genes associated with iCAF identity, and those associated with the SASP, however, this is rarely stated clearly in the CAF literature and most studies have not taken this potential overlap into account. Nevertheless, we argue that the points made in Figure 2 are novel in light of the following points:

1. The vast majority of senescence studies have been conducted on fibroblasts induced to undergo senescence in culture, and there have been few studies to isolate and profile endogenously formed senescent CAFs from tumor lesions. This is in fact the first study to our knowledge to directly isolate matched senescent and non-senescent CAFs from human tumors and characterize their transcriptome (Figure 2c-f). This isolation relies on the SA-bGal stain, successfully used also in the mice. As such this is among the first demonstrations of senescent cell isolation profiling derived directly from human tissue.

2. In the scRNA-seq analysis (Fig. 2j,l, Fig. S4) we intentionally used a senescence signature that does not rely on SASP cytokines, but, rather on CDK inhibitor genes and cell cycle genes, in order to avoid detecting cells with the same markers. This, to allow detection of senescent cells independently of the SASP, and avoid a biased association of senescence with the iCAF signature. We added this note in the text (p. 10 rows 12-13)

3. The data is also novel in that it provides a first direct comparison of datasets from mice, cultured models, patients, and scRNA seq, thereby presenting a comprehensive detailed description of the expression profiles of tumor-derived senescent CAFs.

- Figure 3a-b â The correlation is not very strong, and given that a majority, but certainly not all senescent cells are CAFs (only ~60% of senescent cells in the TME are CAFs), the authors should support the correlation between senescent CAFs and CD3 staining by costaining for CAF markers (i.e correlate PDPN+P16+ cells with presence of CD3+ cells) in order to prove that the effect seen due to senescence is really coming from CAFs and not from other cells. This is particularly important since in Fig 3i the authors show that the total number of CD3+ cells does not change following senescence activation in the injected CAF system.

We address this important question in the revised version. Firstly, we provide a more detailed analysis of the composition of p16+ stromal cells in the KPC model. We show that ~70% of cells are CAFs, whereas other p16+ cell types are much rarer (see reply for next question). This is shown in new Figure 3a,b. We also revised the correlation analysis, and show a negative correlation between the percentage of Pdpn+ CAFs that are p16+, and the numbers of CD3+ T cells per tumor area. In contrast, the overall CAF content does not negatively correlate with CD3+ T cell numbers. This is shown in revised Figure 3d,e, and discussed in p.11 rows 12-17. Note that in these correlation analyses we are limited by the number of individual monoclonal KPC tumors â a total of 9 that could be analyzed.

Following up on this point, this is minor, however what is the other population expressing senescent markers in this model (the remaining 30% in Supp Figure B)? This could potentially be another stromal population (pericytes? Endothelial cells? The authors do not mention this) that could induce similar effects.

As noted above we now provide a more detailed analysis of the composition of the p16+ stromal population in the KPC model (6422c cells). This in fact required a major effort, due to the need to identify compatible antibodies that could be co-stained with p16. We found that ~70% of p16+ cells are indeed Pdpn+ CAFs, 9% are Cd31+ endothelial cells, and 8% are Cd45+ immune cells, approximately half of which are Mac2+ macrophages. CAFs are therefore the major senescent stromal population. This data is now shown as Figure 3a,b, and noted in the text (p.11 rows 8-12).

- Following up on the previous point, in Figure 4 once again the model removes all senescent cells in the mouse, not just the senescent fibroblasts, so the results may be due to other cells. Moreover, the effects are modest and no difference in tumor progression is shown.

This is indeed an important point. In addition to showing that the Pdpn+ subpopulation represents the great majority of the senescent stromal cells as noted above, we now added stains and a quantification demonstrating that the percentage of p16+ CAFs (Pdpn+) is reduced in the two elimination systems used. This is shown for the Ink-ATTAC model in Figure 5b (bottom panel) and Figure 5d, and for the ABT-199 senolytic elimination in Figure 6b (bottom panel) and Figure 6d (these experiments were presented as Figure 4 in the original submission). This is consistent with senescent CAF removal playing the major role in the observed effect on the TME. However, we note that we cannot rule out a role for other stromal senescent cells in the Discussion, p. 19 rows 13-18.

While in some of the experiments we found that senescent cell elimination caused a trend towards reduced tumor growth (see for example Figure 5j, and data not shown), we do state that removal of stromal CAFs is not sufficient to substantially retard tumor growth in our experiments (p.14,15). The effect of senescent cell removal is therefore more strongly observed when combined with the immunotherapy.

- Figure 5 - I appreciate the proof of concept of the combination of the senolytic drug with ICB therapy, however I wonder whether this harsh therapeutic combination is durable and well tolerated.

The senolytic treatment did not have adverse effects on the mice, when delivered on its own, or in combination with the immunotherapy. In this model, the main health burden on the mice are the rapidly-growing pancreatic tumors. Therapy effects were therefore beneficial to the mice, as they dramatically reduced tumor burden. Treatments were administered over 2 weeks out of 3 weeks of tumor growth. Whether such combinations are translatable to human patients remains to be tested in clinical trials. ABT-199 is currently administered to CLL and AML patients as part of therapeutic combinations and is overall well tolerated, yet a combination with specific immunotherapies is yet to be tested.

- The authors should perform mass spectrometry on the CM of the senescent CAFs in order to unravel the mechanisms behind the effect they have on T cells activation.

Following this suggestion we conducted mass spectrometry analysis of the conditioned media of control and senescent human CAFs. This provides an interesting view of upregulated and downregulated components in the secretome, pointing to several secreted factors similarly observed in the CAF transcriptomes. This new data is now presented in Figures S3h. Unraveling the molecular interaction mechanisms will require, however, much further followup work, as there are multiple candidate players detected in the transcriptomes and proteome each with complex effects to be dissected.

- The authors should stain human samples for senescent CAF markers together with markers of CD8+ T cells to validate that the observed phenotype has human relevance.

We now added an analysis of human PDAC sections addressing this question, shown in new Figure 3f,g and described in p.11-12. We calculated the percentages of CAFs expressing p16 in 57 tumor regions in 11 patients, and compared those to the numbers of CD3+, CD8+, and GZMB+CD8+ T cells in each region. Interestingly, a negative correlation with CAF senescence was observed most strongly in activated CD8+ T cells. These findings are consistent with our mouse observations. The high level of variability in human PDAC TME architecture will no doubt require more

detailed analyses of multiple markers on larger cohorts in future studies.

Minor

- Figure 1c - what are the numbers on top of the bars of the graph? It is not mentioned in the figure legend and it doesn't seem like the average value.

We thank the reviewer for noting this typo, these values (averages) were accidentally switched and the error has been fixed.

- Figure 1d - row 19 page 6 "Stromal p16 expression was often observed in regions surrounding the epithelial lesions, and the majority of p16+ cells co-localized with expression of the CAF marker Pdpn" - the sentence needs to be supported by quantification: number of samples and measure of proximity or colocalization.

We now added these quantifications as noted above, see panels Figure 1e,f.

- Figure 1e - quantification is missing.

This panel is now Figure 1g, see new quantification panel added as Figure 1h.

- Figure 1f - the colors of the writing don't match the colors of the image and it can be confusing for some readers.

This mismatch was due to the overlap of green and white signal. The marking is now corrected and a quantification was added to this panel (now Figure 1i,j).

- Figure 1i - the number of replicates is not specified.

This was corrected (now Figure 1m).

- Row 20 page 7: "Most sections containing human premalignant lesions showed the presence of p16+ cells in the epithelial as well as stromal compartments (8 of 17)" - The authors should rephrase this sentence since 8 of 17 is not "most".

This text has been corrected, and the data presentation revised, now shown as Figure 1n.

- Supplementary Figure 1d - what is the threshold to define the area positive or negative for p16+? It should be elaborated in the methods or in the figure legend.

This information is now detailed in the Methods section.

- The authors may wish to reconsider the organization of figure 2: instead of having in the same figure PanIN mouse and PDAC human perhaps insert in the main figure the RNAseq of PDAC mouse (switch between supplementary and main).

We thank the reviewer for this suggestion. However, since the PanIN CAF transcriptomes were of somewhat higher technical quality we decided to leave this data in the main figure.

- Supplementary figure 4c-d: it is claimed that the p16+ stromal regions trend towards having lower CD3+ content, but the representative images shown are misleading since they contain a large amount of cancer areas.

These images were replaced with more appropriate images, now Supplementary Figure 5b,c.

- Figure 3a: scale bars are not defined. The four pictures seem in different magnifications.

All scale bars represent 20 microns. This is now more clearly stated in the legend, and additional scale bars were added to the images (now Figure 3c). The fluorescent images in this panel are in a slightly higher magnification than the histochemistry images.

- Page 13 row 9 "senescent cell elimination is restricted to the stroma" - it is restricted in all cells of the mouse, not just in the stroma. I suggest rephrasing.

This text has been corrected (now p.14 rows 18-20)

- Figure 5b - increase quality of the image.

Unfortunately the PDF conversion process of the manuscript submission system reduces image resolution â high quality images will be included in the published version.

Reviewer #3 (expert in cancer and senescence):

This is an excellent paper characterizing the role of senescent CAFs in the immunogenicity of pancreatic cancers. I basically just want to congratulate the authors for this complete, original, elegant and relevant investigation.

We thank the reviewer for this highly supportive assessment.

I donât really have any criticism, but a couple of curiosities that authors may want to consider:

1. Fig. 3b. This figure shows a negative correlation between p16+ CAFs and intratumoral CD8. I wonder if this negative correlation is specific for p16+ CAFs or if it also true for p16- CAFs or for CAFs in general.

We revised this analysis to address this question. We now show a negative correlation between the percentage of Pdpn+ CAFs that are p16+, and the numbers of CD3+ T cells per tumor area. In contrast, the overall CAF content does not negatively correlate with CD3+ T cell numbers. See revised Figure 3d,e (previously Figure 3a,b). This establishes that indeed the correlation is specific to p16+ CAFs. Note that as in the original manuscript, the correlation here is relative to CD3+ T cells. There is in fact a similar trend of correlation also to CD8+ T cells and Gzmb+CD8+ T cells, yet it does not pass statistical significance due to the small n number of tumors that can be tested in this panel, and therefore is shown as Supplementary Fig. 5a.

2. Figs. 3j,k,l: These figures demonstrate the inhibitory effect of the SASP on CD8 cells in vitro. I wonder what would be the effect if CAFs and CD8 cells are able to interact? This would recapitulate closer the situation in vivo. Senescent cells have been reported to express immunosuppressive and immunostimulatory cell surface ligands.

To address this question we conducted experiments in which we co-cultured human T cells with control and senescent human CAFs. As opposed to the conditioned media experiments, we could not detect an inhibition of the CD8+ T cells. This does not support the suggested interesting idea that the two cell types interact through cell surface ligands. However, since the co-culture experiments are more complex technically, as they require conditions that fit both cell types, we deem this result inconclusive at this point, and therefore did not include this information in the manuscript.

3. I guess a prime candidate factor from the SASP to mediate the immunosuppression of CD8 responses is TGF β (which they also find upregulated in their senescent CAFs). A comment on this would be appropriate. Have authors tried galunisertib in their in vitro assays? (this does not seem too complicated and it would be a nice addition to their already very complete paper).

This is a good suggestion, and we re-tested in this period the potential role of the TGF-beta pathway. However, our data do not demonstrate that TGF-beta ligands are consistently upregulated or secreted in the senescent CAFs, and therefore this pathway does not seem to be the central player in this case. Our transcriptomic and secretomic data hold several mechanistic candidates, yet testing them individually will require much further work, and the overall effect in vivo may be mediated by a combination of factors.

4. Regarding Fig. S7: I found interesting the high levels of BclXL in the tumoral compartment and the depletion of p16+ cancer cells by navitoclax. Perhaps this is worth mentioning.

We added text referring to this information, which is now presented as Fig. S9 (p.15, rows 20-23).

Version 1:

Reviewer comments:

Reviewer #1

(Remarks to the Author)

The authors have thoughtfully and meaningfully addressed my comments from the previous submission.

Reviewer #2

(Remarks to the Author)

The authors have nicely addressed all of my comments.

One minor suggestion: in response to my comment 3 the authors performed staining and quantification of the percentage of p16 positive cell out of VIM+ cells, however, the staining is not shown. I suggest including the staining of VIM as well.

Reviewer #3

(Remarks to the Author)

Thanks for your clarifications. I think this is an excellent paper.
Congratulations.

Author Rebuttal letter:

Response to Reviewers:

We thank the three reviewers for their constructive and positive comments throughout this process.

Reviewer #1 (Remarks to the Author):

The authors have thoughtfully and meaningfully addressed my comments from the previous submission.

Reviewer #2 (Remarks to the Author):

The authors have nicely addressed all of my comments.

One minor suggestion: in response to my comment 3 the authors performed staining and quantification of the percentage of p16 positive cell out of VIM+ cells, however, the staining is not shown. I suggest including the staining of VIM as well.

We believe the reviewer is referring to the data quantification of p16+ cells out of CAFs, which was introduced in the revised version, and now shown in Figure 4d. Here actually it was Pdpn that was used rather than Vim, as the CAF counterstain. We did not include additional images specifically for this added correlation analysis, because the revised version added similar co-stained images of p16 and Pdpn in Figures 4a, 6b and 7b. For this reason adding further images in Figure 4 to those already shown in panels 4a and 4c, seemed somewhat redundant.

Reviewer #3 (Remarks to the Author):

Thanks for your clarifications. I think this is an excellent paper.
Congratulations.
